# Wide Neural Networks as Gaussian Processes: Lessons from Deep Equilibrium Models

**Tianxiang Gao**[*]
Iowa State University
gaotx@iastate.edu

**Xiaokai Huo**[*]
Iowa State University
xhuo@iastate.edu

**Hailiang Liu**
Iowa State University
hliu@iastate.edu

**Hongyang Gao**
Iowa State University
hygao@iastate.edu

## Abstract

Neural networks with wide layers have attracted significant attention due to their equivalence to Gaussian processes, enabling perfect fitting of training data while maintaining generalization performance, known as benign overfitting. However, existing results mainly focus on shallow or finite-depth networks, necessitating a comprehensive analysis of wide neural networks with infinite-depth layers, such as neural ordinary differential equations (ODEs) and deep equilibrium models (DEQs). In this paper, we specifically investigate the deep equilibrium model (DEQ), an infinite-depth neural network with shared weight matrices across layers. Our analysis reveals that as the width of DEQ layers approaches infinity, it converges to a Gaussian process, establishing what is known as the Neural Network and Gaussian Process (NNGP) correspondence. Remarkably, this convergence holds even when the limits of depth and width are interchanged, which is not observed in typical infinite-depth Multilayer Perceptron (MLP) networks. Furthermore, we demonstrate that the associated Gaussian vector remains non-degenerate for any pairwise distinct input data, ensuring a strictly positive smallest eigenvalue of the corresponding kernel matrix using the NNGP kernel. These findings serve as fundamental elements for studying the training and generalization of DEQs, laying the groundwork for future research in this area.

## 1 Introduction

Neural networks with wide layers have recently received significant attention due to their intriguing equivalence to Gaussian processes, known as the Neural Network and Gaussian Process (NNGP) correspondence. It has been established that two-layer fully-connected networks tend towards Gaussian processes as the width of the layers approaches infinity [29, 25]. This equivalence has also been theoretically demonstrated in various neural network architectures, including deep feed-forward networks [28], convolutional neural networks [33, 16], recurrent networks [39], and residual neural networks [35]. This equivalence not only sheds light on the training dynamics of these networks but also highlights their generalization performance, especially when the corresponding covariance matrix is strictly positive definite. These discoveries have paved the way for overparameterized neural networks to achieve perfect fit on training data [11, 31, 3], while maintaining low generalization error on unseen data [4, 30, 2], a phenomenon known as benign overfitting [7, 8, 27].

---

[*]Both authors contributed equally.

37th Conference on Neural Information Processing Systems (NeurIPS 2023).

In recent years, the emergence of infinite-depth neural network architectures, such as neural ordinary differential equations (ODEs) [9] and deep equilibrium models (DEQs) [6], has demonstrated their potential to capture complex dynamic behaviors and achieve superior modeling capabilities [24, 34, 22, 6]. However, the analysis of these architectures in the context of wide neural networks with infinite-depth layers remains largely unexplored. Understanding the convergence properties and relationship to Gaussian processes of these networks is crucial to unravel their underlying mechanisms and unlock their full potential. While some limited studies have investigated the convergence properties of neural ODEs or ResNet architectures [21], such as demonstrating the convergence to a diffusion process in the infinite-depth limit for a specific ResNet architecture [35] and introducing scaling to allow the interchange of the two limits [20], to the best of our knowledge, there is no existing work studying the commutative limits of DEQs.

In this paper, we focus on analyzing the deep equilibrium model (DEQ), an infinite-depth neural network architecture with shared weight matrices across layers. Our objective is to comprehensively analyze the properties of DEQs in the context of wide neural networks and investigate their convergence behavior as the width of the layers tends to infinity. We establish that as the width approaches infinity, DEQ tends to a Gaussian process. Furthermore, under appropriate scaling of the weight matrices, the limits of depth and width commute, enhancing our understanding of the convergence properties of DEQs. Additionally, we demonstrate that the resulting covariance function is strictly positive definite for any distinct input data, provided that the activation function is non-polynomial.

## 2    Related Works

Implicit neural networks [12], such as deep equilibrium models (DEQs), have gained significant attention in the research community over the past decade. Recent studies have shown that implicit neural network architecture encompasses a broader class of models, making it a versatile framework that includes feed-forward neural networks, convolutional neural networks, residual networks, and recurrent neural networks [12, 6]. Moreover, DEQs have been recognized for their competitive performance compared to standard deep neural networks, offering the advantage of achieving comparable results while demanding much fewer computational and memory resources, especially due to the utilization of shared weights [6]. Despite the practical success of DEQs in various real-world applications, our theoretical understanding of DEQs remains limited.

On the other hand, numerous studies [29, 25, 28, 33, 16, 39, 35] have made observations that finite-depth neural networks with random initialization tend to exhibit behavior similar to Gaussian processes as the width approaches infinity, known as NNGP correspondence. This correspondence has led to investigations into the global convergence properties of gradient-based optimization methods. The work of [23] established that the trajectory of the gradient-based method can be characterized by the spectral property of a kernel matrix that is computed by the so-called neural tangent kernel (NTK). Consequently, if the limiting covariance function or NNGP kernel $\Sigma^L$ can be shown to be strictly positive definite under mild conditions, simple first-order methods such as stochastic gradient descent can be proven to converge to a global minimum at a linear rate, provided the neural networks are sufficiently overparameterized [11, 3, 40, 32, 5, 31, 15, 13]. Furthermore, this equivalence offers valuable insights into the generalization performance of neural networks on unseen data. It suggests that wide neural networks can be viewed as kernel methods, and the Rademacher complexity of these networks can be easily computed if the parameter values remain bounded during training. As a result, a line of current research [4, 30, 2, 7, 8, 27, 14] has demonstrated that gradient-based methods can train neural networks of various architectures to achieve arbitrarily small generalization error, given that the neural networks are sufficiently overparameterized.

Unfortunately, the existing results in the literature primarily focus on finite-depth neural networks, and there has been relatively limited research investigating the training and generalization properties of infinite-depth neural networks. One key challenge arises when the limits of depth and width do not commute, leading to distinct behaviors depending on whether the depth or width is relatively larger. For instance, [35] demonstrates that a ResNet with bounded width tends to exhibit a diffusion process, while [20] observes heavy-tail distributions in standard MLPs when the depth is relatively larger than the width. These unstable behaviors give rise to a loss of expressivity in large-depth neural networks, specifically in terms of perfect correlation among network outputs for different inputs, as highlighted by studies such as [36, 37, 19]. This raises significant issue, as it indicates the networks loss the covariance structure from the inputs as growth of depth. In the case of ResNets, a proposed solution

to mitigate this problem involves employing a carefully chosen scaling on the residual branches [20, 18], resulting in commutative limits of depth and width. However, to the best of our knowledge, there is currently no research exploring the interplay between the limits of depth and width for DEQs, which represent another class of infinite-depth neural networks with shared weights.

## 3  Preliminary and Overview of Results

In this paper, we consider a simple deep equilibrium model $f_\theta(x)$ defined as follows:

$$f_\theta(x) = V^T h^*(x). \tag{1}$$

Here, $h^0(x) = 0$, and $h^*(x)$ represents the limit of the transition defined by:

$$h^\ell(x) = \phi(W^T h^{\ell-1}(x) + U^T x), \tag{2}$$

$$h^*(x) = \lim_{\ell \to \infty} h^\ell(x), \tag{3}$$

where $\phi(\cdot)$ is an activation function. The parameters are defined as $U \in \mathbb{R}^{n_{in} \times n}$, $W \in \mathbb{R}^{n \times n}$, and $V \in \mathbb{R}^{n \times n_{out}}$. The following equilibrium equality arises from the fact that $h^*(x)$ is a fixed point of the equation (2):

$$h^*(x) = \phi(W^T h^*(x) + U^T x). \tag{4}$$

To initialize the parameters $\theta =: \text{vec}(U, W, V)$, we use random initialization as follows:

$$U_{ij} \overset{\text{iid}}{\sim} \mathcal{N}\left(0, \frac{\sigma_u^2}{n_{in}}\right), \quad W_{ij} \overset{\text{iid}}{\sim} \mathcal{N}\left(0, \frac{\sigma_w^2}{n}\right), \quad V_{ij} \overset{\text{iid}}{\sim} \mathcal{N}\left(0, \frac{\sigma_v^2}{n}\right), \tag{5}$$

where $\sigma_u, \sigma_w, \sigma_v > 0$ are fixed variance parameters.

To ensure the well-definedness of the neural network parameterized by (1), we establish sufficient conditions for the existence of the unique limit $h^*$ by leveraging fundamental results from random matrix theory applied to random square matrices $A \in \mathbb{R}^{n \times n}$:

$$\lim_{n \to \infty} \frac{\|A\|_{op}}{\sqrt{n}} = \sqrt{2} \quad \text{almost surely (a.s.),}$$

where $\|A\|_{op}$ denotes the operator norm of $A$.

**Proposition 3.1** (Informal Version of Lemma F.1). *There exists an absolute small constant $\sigma_w > 0$ such that $h^*(x)$ is uniquely determined almost surely for all $x$.*

While similar results have been obtained in [15] using non-asymptotic analysis, our contribution lies in the asymptotic analysis, which is essential for studying the behaviors of DEQ under the limit of width approaching infinity. This asymptotic perspective allows us to investigate the convergence properties and relationship between the limits of depth and width. We refer readers to Section 4 and Theorem 4.4 where we leverage this result to demonstrate the limits of depth and width commutes.

After ensuring the well-definedness of $h^*(x)$, the next aspect of interest is understanding the behavior of the neural network $f_\theta$ as a random function at the initialization. Previous studies [28, 33, 16, 39, 35] have demonstrated that finite-depth neural networks behave as Gaussian processes when their width $n$ is sufficiently large. This raises the following question:

> *Q1: Do wide neural networks still exhibit Gaussian process behavior when they have infinite-depth, particularly with shared weights?*

Unfortunately, the answer is generally *No*. The challenge arises when the limits of depth and width do not commute. Several studies have observed that switching the convergence sequence of depth and width leads to different limiting behaviors. While wide neural networks behave as Gaussian processes, [26, 20] have observed heavy-tail distributions when the depth becomes relatively larger than the width. However, in the case of DEQs, we demonstrate that such deviations from Gaussian process behavior do not occur, since the infinite width limit and infinite depth limit do commute for DEQs given by (2). This crucial property is established through a meticulous analysis, focusing on fine-grained analysis to accurately determine the convergence rates of the two limits. Our findings affirm the stability and consistent Gaussian process behavior exhibited by DEQs, reinforcing their unique characteristics in comparison to other wide neural networks with infinite-depth layers.

**Theorem 3.1** (Informal Version of Theorem 4.4). *Under the limit of width $n \to \infty$, the neural network $f_\theta$ defined on (1) tends to a centered Gaussian process with a covariance function $\Sigma^*$.*

Once we have confirm width DEQs acts as Gaussian process, given a set of inputs, the corresponding multidimensional Gaussian random vectors are of interest, especially the nondegeneracy of the covariance matrix. This raises the following question:

*Q2: Is the covariance function $\Sigma^*$ strictly positive definite?*

If the covariance function $\Sigma^*$ of the Gaussian process associated with the DEQ is strictly positive definite, it implies that the corresponding covariance matrix is nondegenerate and has a strictly positive least eigenvalue. This property is crucial in various classical statistical analyses, including inference, prediction, and parameter estimation. Furthermore, the strict positive definiteness of $\Sigma^*$ has implications for the global convergence of gradient-based optimization methods used in training neural networks. In the context of wide neural networks, these networks can be viewed as kernel methods under gradient descent, utilizing the NTK [23]. By making appropriate assumptions on the activation function $\phi$, we establish that the covariance function $\Sigma^*$ of DEQs is indeed strictly positive definite, meaning that the corresponding covariance matrix $K^*$ has strictly positive least eigenvalue when the inputs are distinct.

**Theorem 3.2** (Informal Version of Theorem 4.5). *If the activation function $\phi$ is nonlinear but non-polynomial, then the covariance function $\Sigma^*$ is strictly positive definite.*

These findings expand the existing literature on the convergence properties of infinite-depth neural networks and pave the way for further investigations into the training and generalization of DEQs.

## 4  Main Results

To study the DEQ, we introduce the concept of finite-depth neural networks, denoted as $f_\theta^L(x) = V^T h^{L-1}(x)$, where $h^\ell(x)$ represents the post-activation values. The definition of $h^\ell(x)$ for $\ell \in [L-1]$ is as follows:

$$
\begin{aligned}
g^1(x) &= U^T x, \quad h^1(x) = \phi(g^1(x)), \\
g^\ell(x) &= W^T h^{\ell-1}, \quad h^\ell(x) = \phi(g^\ell(x) + g^1(x)), \text{ for } \ell = 2, 3, \ldots, L-1.
\end{aligned}
\tag{6}
$$

**Remark 4.1.** *We assume $U$, $W$, and $V$ are randomly initialized according to (5). The post-activation values $h^\ell$ differ slightly from those in classical Multilayer Perceptron (MLP) models due to the inclusion of input injection. It is worth mentioning that $f_\theta^L$ is equivalent to the DEQ $f_\theta$ when we let $L \to \infty$, provided that the limit exists.*

### 4.1  $f_\theta^L$ as a Gaussian Process

The finite-depth neural network $f_\theta^L$ can be expressed as a Tensor Program, which is a computational algorithm introduced in [39] for implementing neural networks. In their work, [39] provides examples of various neural network architectures represented as tensor programs. They also establish that all G-var (or pre-activation vectors in our case) in a tensor program tend to Gaussian random variables as the width $n$ approaches infinity [39, Theorem 5.4]. Building upon this result, we can employ a similar argument to demonstrate that the neural network $f_\theta^L$ defined by (6) converges to a Gaussian process, with the covariance function computed recursively, under the assumption of a controllable activation function.

**Definition 4.1.** *A real-valued function $\phi : \mathbb{R}^k \to \mathbb{R}$ is called **controllable** if there exists some absolute constants $C, c > 0$ such that $|\phi(x)| \le C e^{c \sum_{i=1}^k |x_i|}$.*

It is important to note that controllable functions are not necessarily smooth, although smooth functions can be easily shown to be controllable. Moreover, controllable functions, as defined in [39, Definition 5.3], can grow faster than exponential but remain $L^1$ and $L^2$-integrable with respect to the Gaussian measure. However, the simplified definition presented here encompasses almost most functions encountered in practice.

Considering the activation function $\phi$ as controllable and conditioned on previous layers, we observe that the pre-activation $g_k^\ell(x)$ behaves like independent and identically distributed (i.i.d.) Gaussian

random variables. Through induction, both the conditioned and unconditioned distributions of $g_k^\ell(x)$ converge to the same Gaussian random variable $z^\ell(x)$ as the limit approaches infinity. This result is proven in Appendix B.

**Theorem 4.1.** *For a finite-depth neural network $f_\theta^L$ defined in (6), as the width $n \to \infty$, the output functions $f_{\theta,k}^L$ for $k \in [1, n_{out}]$ tends to centered Gaussian processes in distribution with covariance function $\Sigma^L$ defined recursively as follows: for all $\ell \in [2, L-1]$*

$$\Sigma^1(x, x') = \sigma_u^2 \langle x, x' \rangle \tag{7}$$

$$\Sigma^2(x, x') = \sigma_w^2 \mathbb{E}\phi(z^1(x))\phi(z^1(x')) \tag{8}$$

$$\Sigma^{\ell+1}(x, x') = \sigma_w^2 \mathbb{E}\phi(z^\ell(x) + z^1(x))\phi(z^\ell(x') + z^1(x')), \tag{9}$$

*where*

$$\begin{bmatrix} z^1(x) \\ z^\ell(x) \\ z^1(x') \\ z^\ell(x') \end{bmatrix} \sim \mathcal{N}\left(0, \left[\begin{array}{cc|cc} \Sigma^1(x,x) & 0 & \Sigma^1(x,x') & 0 \\ 0 & \Sigma^\ell(x,x) & 0 & \Sigma^\ell(x,x') \\ \hline \Sigma^1(x',x) & 0 & \Sigma^1(x',x') & 0 \\ 0 & \Sigma^\ell(x',x) & 0 & \Sigma^\ell(x',x') \end{array}\right]\right) \tag{10}$$

Furthermore, we derive a compact form of the covariance function $\Sigma^L$ in Corollary 4.2 by using the fact $z^1$ and $z^\ell$ are independent, which is proven in Appendix C.

**Corollary 4.2.** *The covariance function $\Sigma^L$ in Theorem 4.1 is rewritten as follows: $\forall \ell \in [1, L-1]$*

$$\Sigma^1(x, x') = \sigma_u^2 \langle x, x' \rangle / n_{in}, \tag{11}$$

$$\Sigma^{\ell+1}(x, x') = \sigma_w^2 \mathbb{E}\phi(u^\ell(x))\phi(u^\ell(x')), \tag{12}$$

*where $(u^\ell(x), u^\ell(x'))$ follows a centered bivariate Gaussian distribution with covariance*

$$\mathrm{Cov}(u^\ell(x), u^\ell(x')) = \begin{cases} \Sigma^1(x, x'), & \ell = 1 \\ \Sigma^\ell(x, x') + \Sigma^1(x, x'), & \ell \in [2, L-1] \end{cases} \tag{13}$$

**Remark 4.2.** *It is worth noting that the same Gaussian process or covariance function $\Sigma^L$ is obtained regardless of whether the same weight matrix $W$ is shared among layers. Additionally, there is no dependence across layers in the limit if different weight matrices are used. That is, if $W^\ell \neq W^k$, then $\mathrm{Cov}(z^\ell(x), z^k(x')) = 0$. These observation align with studies of recurrent neural networks [1, 39], where the same weight matrix $W$ is applied in each layer.*

## 4.2 On the Strictly Positive Definiteness of $\Sigma^L$

To clarify the mathematical context, we provide a precise definition of the strict positive definiteness of a kernel function:

**Definition 4.2.** *A kernel function $k : X \times X \to \mathbb{R}$ is said to be strictly positive definite if, for any finite set of pairwise distinct points $x_1, x_2, \ldots, x_n \in X$, the matrix $K = [k(x_i, x_j)]_{i,j=1}^n$ is strictly positive definite. In other words, for any non-zero vector $c \in \mathbb{R}^n$, we have $c^T K c > 0$.*

Recent works [11, 31, 38, 3] have studied the convergence of (stochastic) gradient descent to global minima when training neural networks. It has been shown that the covariance function or NNGP kernel $\Sigma^L$ being strictly positive definite guarantees convergence. In the case where the data set is supported on a sphere, we can establish the strict positive definiteness of $\Sigma^L$ using Gaussian integration techniques and the existence of strictly positive definiteness of priors. The following theorem (Theorem 4.3) is proven in Appendix D.

**Theorem 4.3.** *For a non-polynomial Lipschitz nonlinear $\phi$, for any input dimension $n_0$, the restriction of the limiting covariance function $\Sigma^L$ to the unit sphere $\mathbb{S}^{n_0-1} = \{x : \|x\| = 1\}$, is strictly positive definite for $2 \leq L < \infty$.*

This theorem establishes that the limiting covariance function $\Sigma^L$ of finite-depth neural network $f_\theta^L$ is strictly positive definite when restricted to the unit sphere $\mathbb{S}^{n_0-1}$, provided that a non-polynomial activation function is used.

### 4.3 $f_\theta$ as a Gaussian Process

In this subsection, we explore the convergence behavior of the infinite-depth neural network $f_\theta$ to a Gaussian process as the width $n$ tends to infinity. Since we have two limits involved, namely the depth and the width, it can be considered as a double sequence. Therefore, it is essential to review the definitions of convergence in double sequences.

**Definition 4.3.** *Let $\{a_{m,n}\}$ be a double sequence, then it has two types of **iterated limits***

$$\lim_{m\to\infty}\lim_{n\to\infty} a_{m,n} = \lim_{m\to\infty}\left(\lim_{n\to\infty} a_{m,n}\right),\tag{14}$$

$$\lim_{n\to\infty}\lim_{m\to\infty} a_{m,n} = \lim_{n\to\infty}\left(\lim_{m\to\infty} a_{m,n}\right).\tag{15}$$

*The **double limit** of $\{a_{m,n}\}$ is denoted by*

$$L := \lim_{m,n\to\infty} a_{m,n},\tag{16}$$

*which means that for all $\varepsilon > 0$, there exists $N(\varepsilon) \in \mathbb{N}$ s.t. $m,n \geq N(\epsilon)$ implies $|a_{m,n} - L| \leq \epsilon$.*

In Subsection 4.1, we have previously shown that $f_\theta^L$ converges to a centered Gaussian process with a covariance function $\Sigma^L$, which is recursively defined. However, it is important to note that this convergence does not necessarily imply that the infinite-depth neural network $f_\theta$ also converges to a Gaussian process, as the order in which the limits are taken can affect the result. Recent studies, such as [28, 20], have demonstrated that the convergence behavior of neural networks depends on the order in which the width and depth limits are taken. Specifically, when the width tends to infinity first, followed by the depth, a standard multi-layer perceptron (MLP) converges weakly to a Gaussian process. However, if the depth tends to infinity first, followed by the width, a heavy-tail distribution emerges. Additionally, when both the depth and width tend to infinity at a fixed ratio, a log-normal distribution is observed for Residual neural networks. Hence, the two limits are not necessarily equivalent unless they commute.

When studying the convergence behaviors of DEQs, it is more crucial to focus on the infinite-depth-then-width limit, as DEQs are defined as infinite-depth neural networks. Therefore, to establish the Gaussian process nature of DEQs, it is important to show that the infinite-depth-then-width limit is equal to the infinite-width-then-depth limit. Fortunately, we can demonstrate that these two limits commute and are equal to the double limit, as the convergence of the depth is much faster than the width, as proven in Appendix F.

**Lemma 4.1.** *Choose $\sigma_w > 0$ small such that $\gamma := 2\sqrt{2}\sigma_w < 1$. Then for every $x, x' \in \mathbb{S}^{n_{in}-1}$, $\lim_{\ell\to\infty}\lim_{n\to\infty}\frac{1}{n}\left\langle h^\ell(x), h^\ell(x')\right\rangle$ and $\lim_{n\to\infty}\lim_{\ell\to\infty}\frac{1}{n}\left\langle h^\ell(x), h^\ell(x')\right\rangle$ exist and equal to $\Sigma^*(x, x')$ a.s., i.e.,*

$$\Sigma^*(x, x') := \lim_{\ell\to\infty}\lim_{n\to\infty} A_{n,\ell} = \lim_{n\to\infty}\lim_{\ell\to\infty} A_{n,\ell} = \lim_{\ell,n\to\infty} A_{n,\ell},\tag{17}$$

*where $A_{n,\ell} := \frac{1}{n}\left\langle h^\ell(x), h^\ell(x')\right\rangle$.*

*Proof.* Proof is provided in Appendix F $\qquad\square$

Lemma 4.1 confirms the two iterated limits of the empirical covariance $\frac{1}{n}\left\langle h_n^\ell(x), h_n^\ell(x')\right\rangle$ of the pre-activation $g_k^\ell$ exist and equal to the double limit $\Sigma^*(x, x')$ for any $x, x' \in \mathbb{S}^{n_{in}-1}$. Consequently, it establishes the commutation of the two limits, implying that the limit-depth-then-width and limit-width-then-depth have the same limit. Building upon this result, we can state Theorem 4.4, which asserts that as the width $n$ of the infinite-depth neural network $f_\theta$ tends to infinity, the output functions $f_{\theta,k}$ converge to independent and identically distributed (i.i.d.) centered Gaussian processes with the covariance function $\Sigma^*(x, x')$. The detailed proofs can be found in Appendix G.

**Theorem 4.4.** *Choose $\sigma_w > 0$ small such that $\gamma := 2\sqrt{2}\sigma_w < 1$. For infinite-depth neural network $f_\theta$ defined on (1), as width $n \to \infty$, the output functions $f_{\theta,k}$ tend to i.i.d. centered Gaussian processes with covariance function $\Sigma^*$ defined by*

$$\Sigma^*(x, x') = \lim_{\ell\to\infty} \Sigma^\ell(x, x'),\tag{18}$$

*where $\Sigma^\ell$ are defined in Theorem 4.1.*

## 4.4 The Strict Positive Definiteness of $\Sigma^*$

We conclude this section by establishing the strictly positive definiteness of the limiting covariance function $\Sigma^*$. Notably, the proof techniques used in Theorem 4.3 are not applicable here, as the strict positive definiteness of $\Sigma^L$ may diminish as $L$ approaches infinity.

Instead, we leverage the inherent properties of $\Sigma^*$ itself and Hermitian expansion of the dual activation $\hat{\phi}$ of $\phi$ [10]. To explore the essential properties of $\Sigma^*$, we perform a fine analysis on the pointwise convergence of the covariance function $\Sigma^L$ for each pair of inputs $(x, x')$.

**Lemma 4.2.** *Choose $\sigma_w > 0$ small for which $\beta := \frac{\sigma_w^2}{2}\mathbb{E}|z|^2|z^2 - 1| < 1$, where $z$ is standard Gaussian random variable. Then for all $x, x' \in \mathbb{S}^{n_{in}-1}$, the function $\Sigma^\ell$ satisfies*

*1. $\Sigma^\ell(x, x) = \Sigma^\ell(x', x')$,*

*2. $\Sigma^\ell(x, x) \leq (1 + 1/\beta)\Sigma^2(x, x)$.*

*Consequently, $\Sigma^*(x, x') = \lim_{\ell \to \infty} \Sigma^\ell(x, x')$ is well-defined and satisfies for all $x, x' \in \mathbb{S}^{n_{in}-1}$*

$$0 < \Sigma^*(x, x) = \Sigma^*(x', x') < \infty.$$

Lemma 4.2, proven in Appendix E, ensures the well-definedness of the limiting covariance function $\Sigma^*$ for all $x, x' \in \mathbb{S}^{n_{in}-1}$ by choosing a small $\sigma_w > 0$. The lemma also guarantees that $\Sigma^*(x, x)$ and $\Sigma^*(x', x')$ are strictly positive, equal, and finite for all $x, x' \in \mathbb{S}^{n_{in}-1}$. These findings are crucial for demonstrating the strict positive definiteness of $\Sigma^*$. Specifically, by leveraging these properties of $\Sigma^*$, we can derive its Hermitian expansion of the limiting kernel $\Sigma^*$.

By utilizing [23, Theorem 3], we establish in Theorem 4.5, as proven in Appendix H, that $\Sigma^*$ is strictly positive definite if $\phi$ is non-polynomial. It is important to note that our analysis can be extended to analyze the covariance or kernel functions induced by neural networks, particularly those that are defined as limits or induced by infinite-depth neural networks. This is because the analysis does not rely on the existence of the positive definiteness of priors. Instead, we examine the intrinsic properties of $\Sigma^*$, which remain independent of the properties of the activation function $\phi$.

**Theorem 4.5.** *For a non-polynomial Lipschitz nonlinear $\phi$, for any input dimension $n_0$, the restriction of the limiting covariance $\Sigma^*$ to the unit sphere $\mathbb{S}^{n_0-1} = \{x : \|x\| = 1\}$, is strictly positive definite.*

# 5 Experimental Results

In this section, we present a series of numerical experiments to validate the theoretical results we have established. Our experiments aim to verify the well-posedness of the fixed point of the transition equation (2). We also investigate whether the DEQ behaves as a Gaussian process when the width is sufficiently large, as stated in our main result, Theorem 4.4. Additionally, we examine the strictly positive definiteness of the limiting covariance function $\Sigma^*$, as established in Theorem 4.5, by computing the smallest eigenvalue of the associated covariance matrix $K^*$. These experiments serve to empirically support our theoretical findings.

## 5.1 Convergence to the fixed point

Proposition 3.1 guarantees the existence of a unique fixed point for the DEQ. To verify this, we conducted simulations using neural networks with the transition equation (2). We plotted the relative error between $h^\ell$ and $h^{\ell+1}$ in Figure 1. As shown in the figure, we observed that the iterations required for convergence to the fixed point were approximately 25 for various widths. This observation aligns with our theoretical findings in Lemma F.1, where the random initialization (5) scales the weight matrix $W$ such that $\|W\|_{op} = \mathcal{O}(\sigma_w)$.

## 5.2 The Gaussian behavior

Theorem 4.4 predicts that the outputs of DEQ tends to follow a Gaussian process as the width approaches infinity. To demonstrate this, we consider a specific DEQ with $n_{in} = 10$ and $n_{out} = 10$, activated by tanh. We analyze the output distributions of 10,000 neural networks. An important

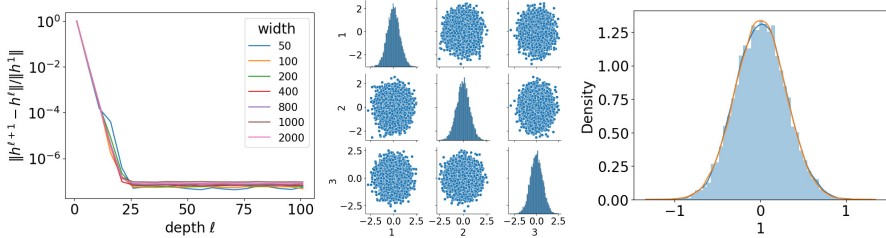

Figure 1: Convergence to the fixed point (left); distribution of the first neuron of the output for 10, 000 neural networks, KS statistics, p-value (middle); joint distributions for the first neuron for three outputs for 10, 000 neuron networks, with orange curve denotes the Gaussian distribution (right)

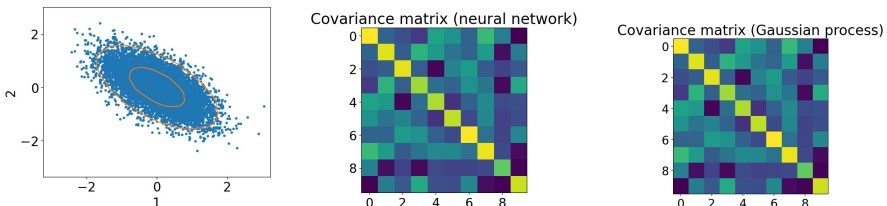

Figure 2: Joint distributions for the first neuron for two different inputs over 10, 000 neural networks (left); Covariance matrix obtained by a neural network (middle); Covariance matrix obtained by Gaussian process (right)

implication of Theorem 4.4 is that the output forms an independent identical Gaussian distribution. To visualize this, we plot a pairplot in Figure 1 illustrating the randomly selected three outputs, confirming the validity of this implication.

Next, we generate histograms of the 10,000 neural networks to approximate the distribution of the first neuron in the output layer. In the third plot of Figure 1, we present the histogram for a width of 1000. Remarkably, the output distribution exhibits a strong adherence to the Gaussian model, as evidenced by a Kolmogorov-Smirnov (KS) statistic of 0.0056 and a corresponding p-value of 0.9136. Furthermore, in Figure 5 in the supplementary material, we provide histograms for widths of 10, 50, 100, 500, and 1000. As the width increases, the output distribution progressively converges towards a Gaussian distribution. This is evident from the decreasing KS statistics and the increasing p-values as the width extends from 10 to 1000.

Based on Theorem 4.4, the outputs of the neural network exhibit a behavior reminiscent of a joint Gaussian distribution for different inputs $x$ and $x'$. To illustrate this, we plot the first output of the 10,000 neural networks for two distinct inputs as the first plot in Figure 2. Notably, the predicted limiting Gaussian level curves, derived from the limiting kernel function stated in Lemma 4.2, perfectly match the results of the simulations when the width is set to 1000.

### 5.3 Convergence of the kernel

According to Theorem 4.4, the DEQs tends to a Gaussian process with a covariance function $\Sigma^* = \lim_{\ell \to \infty} \Sigma^\ell$. Given $N$ distinct inputs $\{x_i\}_{i=1}^N$, as stated in Theorem 4.1, the limiting covariance matrix $K^*$ can be computed recursively, i.e., $K_{ij}^\ell = \Sigma^\ell(x_i, x_j)$. By Lemma 4.1, each element $K_{ij}^\ell$ can be approximated by $\frac{1}{n} \langle h^\ell(x_i), h^\ell(x_j) \rangle$. We conduct a series of numerical experiments to visually assess this convergence

First of all, we examine the convergence in width. We fix a large depth $\ell$ and vary the widths by $2^{2-13}$. We draw the errors between limiting covariance matrix $K^*$ and finite-width empirical estimate $K_n^\ell$ in the first two plots of Figure 3. The relative errors $\|K_n^\ell - K^*\|_F / \|K^*\|_F$ consistently decreases as the growth of the width, and a convergence rate of order $n^{-1}$ is observed.

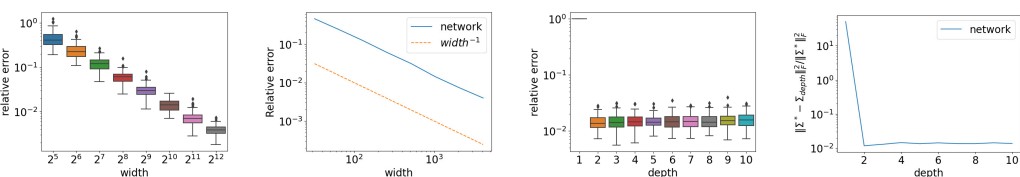

Figure 3: Covariance behaviors with varying width and depth

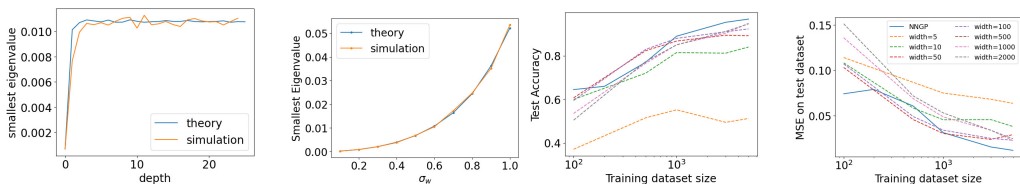

Figure 4: From left to right: $\lambda_{\min}(K^\ell)$ across varying depths $\ell$; $\lambda_{\min}(K^*)$ for different $\sigma_w$ (blue curve: theory; orange curve: simulation); Test accuracy of the MNIST dataset using NNGP and DEQs with various widths; MSE of the MNIST dataset using NNGP and DEQs with various widths.

Next, we examine the convergence in depth by fixing a large width. The results are shown in the third and fourth plots of Figure 3. From these plots, we can observe that the error converges rapidly as the depth of the network increases, illustrating an exponential convergence rate.

### 5.4 The positive definiteness of the kernel

Theorem 4.5 establishes that the NNGP kernel is strictly positive. As discussed earlier, the kernel matrix $K^L$ can be computed recursively, as stated in Theorem 4.1 or Corollary 4.2. We refer to this computation as the *theoretical* approach. Alternatively, it can be calculated as the covariance through simulation, which we denote as *simulation* approach. We employ both methods to compute the smallest eigenvalue of the kernel matrix $K^L$. The results are summarized in Figure 4. It is evident from the figure that the smallest eigenvalues increase with increasing depths and become stable once the kernel is well approximated. Furthermore, the smallest eigenvalue increases with higher values of $\sigma_w$.

### 5.5 Test Performance

To complement the theoretical analysis, we conducted numerical experiments demonstrating the NNGP correspondence for DEQs on real datasets with varying widths. A visual representation of these findings is available in Figure 4. Intriguingly, our observations consistently reveal that the NNGP continually outperforms trained finite-width DEQs. Moreover, a compelling trend emerges: as network width increases, the performance of DEQs converges more closely to NNGP performance. Notably, this phenomenon mirrors observations made in the context of standard feedforward neural networks [25, 28]. These experiments stand as practical evidence, effectively shedding light on the behavior of DEQs across different network sizes. The insights gleaned from these experiments have been thoughtfully integrated into our paper to enhance its comprehensiveness and practical relevance.

## 6 Conclusion and Future Work

This paper establishes that DEQs (Deep Equilibrium Models) can be characterized as Gaussian processes with a strict positive definite covariance function $\Sigma^*$ in the limit of the width of the network approaching infinity. This finding contributes to the understanding of the convergence properties of infinite-depth neural networks, demonstrating that when and how the depth and width limits commute. An important direction for future research is to leverage the results presented in this paper to investigate the training and generalization performance of DEQs. While the results obtained in this paper hold for commonly used activation functions, it would be interesting to explore more complex transition functions in future work.

# 7 Acknowledgements

We would like to acknowledge the generous support of the National Science Foundation (NSF) under grant DMS-1812666 and III-2104797.

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

# A  Useful Mathematical Results

**Theorem A.1.** *Let $A$ be $m \times m$ random matrix whose entries $A_{ij}$ are independent identically distributed standard Gaussian random variables. Then, there exists absolute constant $c, C > 0$ such that*

$$\|A\|_{op} \leq C\sqrt{m}, \quad \text{with probability at least } 1 - 2e^{-cm}. \tag{19}$$

**Theorem A.2** (Strong Bai-Yin theorem)**.** *Let $A$ be $m \times m$ random matrix whose entries $A_{ij}$ are independent identically distributed standard Gaussian random variables. Then*

$$\lim_{m \to \infty} \|A\|_{op}/\sqrt{m} = \sqrt{2}, \quad \text{almost surely.} \tag{20}$$

**Theorem A.3** (Kolmogorov's SLLN for *i.i.d.*)**.** *Let $\{X_n\}$ be sequence of i.i.d. random variables and $S_n = \sum_{i=1}^{n} X_i$. Then $\frac{S_n}{n} \overset{a.s.}{\to} \mathbb{E}X_1$ if and only if $\mathbb{E}|X_1| < \infty$.*

**Lemma A.1** (Almost surely convergence)**.** *Some important properties of almost surely convergence.*

1. *If $X_n \overset{a.s.}{\to} X$, then $g(X_n) \overset{a.s.}{\to} g(X)$ for all continuous function g.*

2. *If $X_n \overset{a.s.}{\to} X$ and $Y_n \overset{a.s.}{\to} Y$, then $X_n Y_n \overset{a.s.}{\to} XY$.*

3. *If $X_n \overset{a.s.}{\to} X$ and $Y_n \overset{a.s.}{\to} Y$, then $aX_n + bY_n \overset{a.s.}{\to} aX + bY$.*

**Lemma A.2** (Gaussian smoothing)**.** *Let $f, g$ be a real-valued function. Define function $F(\sigma) := \mathbb{E}_{z \sim \mathcal{N}(\mu, \sigma^2)} f(z)$ and $G(\mu) = \mathbb{E}_{z \sim \mathcal{N}(\mu, \sigma^2)} g(z)$ for $\sigma > 0$. Suppose $f(x), g(x) \in o(e^{-x^2})$, then*

$$F'(\sigma) = \frac{1}{\sigma} \mathbb{E}_{z \sim \mathcal{N}(0,1)} \left[ f(\mu + \sigma z)(z^2 - 1) \right]$$

$$G'(\mu) = \frac{1}{\sigma} \mathbb{E}_{z \sim \mathcal{N}(0,1)} \left[ g(\mu + \sigma z)z \right]$$

*Proof.* Note that $F(\sigma) = \mathbb{E}_{z \sim \mathcal{N}(0,1)} f(\mu + \sigma z)$, then

$$
\begin{aligned}
F'(\sigma) =& \frac{d}{d\sigma} \int_{-\infty}^{\infty} f(\mu + \sigma z) \frac{1}{\sqrt{2\pi}} e^{-z^2/2} dz \\
=& \int_{-\infty}^{\infty} f'(\mu + \sigma z) z \frac{1}{\sqrt{2\pi}} e^{-z^2/2} dz \\
=& \int_{-\infty}^{\infty} f'(u) \left( \frac{u - \mu}{\sigma} \right) \frac{1}{\sigma\sqrt{2\pi}} e^{-\frac{(u-\mu)^2}{2\sigma^2}} du, \quad u = \mu + \sigma z \\
=& \frac{1}{\sigma} \int_{-\infty}^{\infty} f(u) \left[ \frac{(u-\mu)^2}{\sigma^2} - 1 \right] \frac{1}{\sigma\sqrt{2\pi}} e^{-\frac{(u-\mu)^2}{2\sigma^2}} du \\
=& \frac{1}{\sigma} \int_{-\infty}^{\infty} f(\mu + \sigma z) \left[ z^2 - 1 \right] \frac{1}{\sqrt{2\pi}} e^{-\frac{z^2}{2}} dz \\
=& \frac{1}{\sigma} \mathbb{E}_{z \sim \mathcal{N}(0,1)} \left[ f(\mu + \sigma z)(z^2 - 1) \right]
\end{aligned}
$$

Similarly, we have

$$
\begin{aligned}
G'(\mu) =& \frac{d}{d\mu} \int_{-\infty}^{\infty} g(\mu + \sigma z) \frac{1}{\sqrt{2\pi}} e^{-z^2/2} dz \\
=& \int_{-\infty}^{\infty} g'(\mu + \sigma z) \frac{1}{\sqrt{2\pi}} e^{-z^2/2} dz \\
=& \int_{-\infty}^{\infty} g'(u) \frac{1}{\sigma\sqrt{2\pi}} e^{-\frac{(u-\mu)^2}{2\sigma^2}} du, \quad u = \mu + \sigma z \\
=& \frac{1}{\sigma} \int_{-\infty}^{\infty} g(u) \left( \frac{u-\mu}{\sigma} \right) \frac{1}{\sigma\sqrt{2\pi}} e^{-\frac{(u-\mu)^2}{2\sigma^2}} du \\
=& \frac{1}{\sigma} \int_{-\infty}^{\infty} g(\mu + \sigma z) z \frac{1}{\sqrt{2\pi}} e^{-\frac{z^2}{2}} dz \\
=& \frac{1}{\sigma} \mathbb{E}_{z \sim \mathcal{N}(0,1)} \left[ g(\mu + \sigma z) z \right]
\end{aligned}
$$

$\square$

**Lemma A.3** (Gaussian conditioning). *Given $G \in \mathbb{R}^{n \times m}$ and $H \in \mathbb{R}^{n \times m}$, let $W \in \mathbb{R}^{n \times n}$ to follow matrix Gaussian distribution, i.e., $W \sim \mathcal{MN}(0, \sigma I_n, \sigma I_n)$ for some $\sigma > 0$, suppose $G = WH$ has feasible solutions. Then the conditional distribution of $W$ given on $G = WH$ is*

$$
W|_{G=WH} \sim \mathcal{MN}(GH^\dagger, I_n, \sigma^2 \Pi\Pi^T).
$$

*where $\Pi = I_n - HH^\dagger$ is the orthogonal projection onto the null$(H^T)$.*

*Proof.* First, we consider the optimization problem

$$
\min_W \frac{1}{2} \|W\|_F^2, \quad s.t. \quad G = WH.
$$

The Lagrange function is given by

$$
L(W, V) = \frac{1}{2} \|W\|_F^2 + \langle V, G - WH \rangle.
$$

The KKT condition implies $\nabla_W L(W, V) = W - VH^T = 0$ and further $W = VH^T$. Since $G = WH$, we have $V = G(H^T H)^\dagger$ and so $W^* = G(H^T H)^\dagger H^T = GH^\dagger$.

Then let $\Pi = I_n - HH^\dagger$ be the orthogonal projection onto the null$(H^T)$. Thus, the conditional distribution of $W$ given $G = WH$ is

$$
W|_{G=WH} = GH^\dagger + \tilde{W}\Pi^T = \mathcal{MN}(GH^\dagger, I_n, \sigma^2 \Pi\Pi^T).
$$

$\square$

**Lemma A.4** (Conditional distribution). *Let $X \sim \mathcal{MN}(M, U, V)$. Partition $X$, $M$, and $V$ such that*

$$
X = \begin{bmatrix} X_1 \\ X_2 \end{bmatrix}, \quad M = \begin{bmatrix} M_1 \\ M_2 \end{bmatrix} \quad U = \begin{bmatrix} U_{11} & U_{12} \\ U_{21} & U_{22} \end{bmatrix}
$$

*where $X_1 \in \mathbb{R}^{m \times p}$. Then*

$$
\begin{aligned}
& X_1 \sim \mathcal{MN}(M_1, U_{11}, V) \\
& X_2|X_1 \sim \mathcal{MN}\left( M_2 + U_{21}U_{11}^{-1}(X_1 - M_2), U_{22} - U_{21}U_{11}^{-1}U_{12}, V \right).
\end{aligned}
$$

*Note, if $U_{21} = 0$, then $X_2|X_1 \sim \mathcal{MN}(M_2, U_{22}, V)$ indicates $X_2$ and $X_1$ are **independent**.*

**Lemma A.5.** *Let $\sigma$ be a L-Lipschitz continuous function. Then $\sigma$ is also a controllable function. In addition, $\phi(x, y) := \sigma(x)\sigma(y)$ is also a controllable function.*

*Proof.* WOLG, we can assume $L = 1$. As $\sigma$ is Lipschitz continuous on its region, there must exists some $x_0$ such that $\sigma(x_0) = c$. Then we have

$$|\sigma(x)| \leq |\sigma(x) - \sigma(x_0)| + |c| \leq |x - x_0| + |c| \leq e^{|c|^{-1}|x-x_0|} \leq e^{|c|^{-1}|x_0|}e^{|c|^{-1}|x|} = C_1 e^{C_2|x|}.$$

Similarly, we have

$$|\phi(x,y)| = |\sigma(x)|\,|\sigma(y)| \leq C_1 e^{C_2(|x|+|y|)}.$$

$\square$

**Lemma A.6.** *Let $f$ be a controllable function. Then for all $\mu \in \mathbb{R}$ and $\sigma \geq 0$, we have*

$$\mathbb{E}_{z \sim \mathcal{N}(\mu,\sigma^2)} |f(z)| \leq 2C_1 e^{C_2|\mu|+C_2^2\sigma^2/2}.$$

*Proof.* Note that

$$
\begin{aligned}
\mathbb{E}_{z \sim \mathcal{N}(\mu,\sigma^2)} |f(z)| ={}& \mathbb{E}_{z \sim \mathcal{N}(0,1)} |f(\sigma z + \mu)| \\
\leq{}& \mathbb{E}_{z \sim \mathcal{N}(0,1)} C_1 e^{C_2(\sigma|z|+|\mu|)} \\
={}& C_1 e^{C_2|\mu|} \mathbb{E}_{z \sim \mathcal{N}(0,1)} e^{C_2\sigma|z|} \\
={}& C_1 e^{|\mu|} \int_{-\infty}^{\infty} e^{C_2\sigma|z|} \frac{1}{\sqrt{2\pi}} e^{-z^2/2} dz \\
={}& C_1 e^{|\mu|} \left[ \int_{-\infty}^{0} e^{-C_2\sigma z} \frac{1}{\sqrt{2\pi}} e^{-z^2/2} dz + \int_{0}^{\infty} e^{C_2\sigma z} \frac{1}{\sqrt{2\pi}} e^{-z^2/2} dz \right] \\
={}& C_1 e^{|\mu|} \left[ \int_{-\infty}^{0} \frac{1}{\sqrt{2\pi}} e^{-\frac{1}{2}(z+C_2\sigma)^2+\frac{C_2^2\sigma^2}{2}} dz + \int_{0}^{\infty} \frac{1}{\sqrt{2\pi}} e^{-\frac{1}{2}(z-C_2\sigma)^2+\frac{C_2^2\sigma^2}{2}} dz \right] \\
\leq{}& 2C_1 e^{C_2|\mu|+C_2^2\sigma^2/2}
\end{aligned}
$$

$\square$

# B    Proof of Theorem 4.1

In this Appendix, we show the preactivation $g_k^\ell$ acts like Gaussian random variable. As a consequence, the finite-depth neural network $f_\theta^L$ tends to a Gaussian process as width $n \to \infty$.

**Lemma B.1.** *Suppose the activation function $\phi$ is nonlinear Lipschitz continuous function. For input $x$, let $g^1, \cdots, g^\ell$ be the resulting pre-activations for $\ell \in [L]$. Then for any $\ell \in [L]$ and for any controllable function $\Phi : \mathbb{R}^\ell \to \mathbb{R}$, we have as $m \to \infty$*

$$\frac{1}{n} \sum_{k=1}^n \Phi(g_k^1, \cdots, g_k^\ell) \xrightarrow{a.s.} \mathbb{E}\left[\Phi(z^1, \cdots, z^\ell)\right], \tag{21}$$

*where $(z^i, z^j) \sim \mathcal{N}(0, \Sigma)$ and the covariance matrix $\Sigma \in \mathbb{R}^{2\times 2}$ are computed recursively as follows*

$$\Sigma(z^1, z^i) = \delta_{1,i}\sigma_u^2 \|x\|^2/n_{in}, \qquad\qquad \forall i \geq 1, \tag{22}$$

$$\Sigma(z^i, z^j) = \sigma_w^2 \mathbb{E}\phi(u^{i-1})\phi(u^{j-1}), \qquad\qquad \forall i \geq 2. \tag{23}$$

*where $u^1 = z^1$ and $u^\ell = z^\ell + z^1$ with covariance*

$$\Sigma(u^1, u^i) = \sigma_u^2 \|x\|^2/n_{in}, \qquad\qquad \forall i \geq 1, \tag{24}$$

$$\Sigma(u^i, u^j) = \Sigma(z^i, z^j) + \Sigma(z^1, z^1), \qquad\qquad \forall i \geq 2. \tag{25}$$

*If, in addition, $W^i$ and $W^j$ are independent, then*

$$\Sigma(z^i, z^j) = 0, \quad \forall i \neq j. \tag{26}$$

**Lemma B.2.** *[39, Theorem 5.4] For any NETSOR program whose weight matrices are random initiated as (5) and all activation functions are controllable. If $g^1, \cdots, g^\ell$ are any G-vars (i.e., pre-activation in our case), then for any controllable function $\Phi : \mathbb{R}^\ell \to \mathbb{R}$, we have*

$$\frac{1}{n} \sum_{k=1}^n \Phi(g_k^1, \cdots, g_k^\ell) \xrightarrow{a.s.} \mathbb{E}_{z \sim \mathcal{N}(\mu, \Sigma)}\Phi(z), \tag{27}$$

*where $z := (z^1, \cdots, z^\ell)$ and $\mu$ and $\Sigma$ can computed by [39, Definition 5.2].*

Intuitively, these two Lemmas indicate that $(g_k^1, \cdots, g_k^\ell)$ acts like a multidimensional Gaussian vector whose covariance can be computed recursively. Lemma B.1 is a special case of Lemma B.2 as Lemma B.1 requires each pre-activation $g^\ell$ encoded same input $x$, while Lemma B.2 does not make such assumption. In fact, the proof techniques are identical, *i.e.*, uses Gaussian conditions and smoothing inductively on previous results. To make the paper self-contained, here we provide a proof for Lemma B.1 where we simplify the proof of [39, Theorem 5.4] in the following subsections by removing so-called *core set*.

## B.1    Proof of Theorem 4.1 by Using Master Theorem B.1 or B.2

Based on Lemma B.1 or B.2, we can immediately obtain the desired result.

For simplicity, we assume $\sigma_\ell = 1$. We prove the desired result by induction. For $L = 1$, we have $f_\theta^L(x) = g^1(x) = W^1 x$ and

$$f_{\theta,k}^1(x) = g_k^1(x) = \langle w_k, x \rangle \overset{i.i.d.}{\sim} \mathcal{N}(0, \|x\|^2/n_{in}).$$

Then we have

$$\hat{\Sigma}^1(x, x') = \text{cov}(f_{\theta,k}^L(x), f_{\theta,k}^L(x')) = \langle x, x' \rangle := \Sigma^1(x, x').$$

For $L = 2$, we have $f_\theta^L(x) = g^2(x) = W^2 h^1(x)$. By condition on $g^1$, we have

$$f_{\theta,k}^2(x) = g_k^2(x) = \langle w_k^2, h^1(x) \rangle \overset{i.i.d.}{\sim} \mathcal{N}(0, \|h^1(x)\|^2/n).$$

Then

$$\hat{\Sigma}^2(x, x') = \langle h^1(x), h^1(x') \rangle / n$$
$$= \langle \phi(g^1(x)), \phi(g^1(x')) \rangle / n$$
$$= \frac{1}{n} \sum_{i=1}^{n} \phi(g_i^1(x)) \phi(g_i^1(x'))$$
$$\overset{a.s.}{\to} \mathbb{E}\phi(z^1(x))\phi(z^2(x'))$$
$$=: \Sigma^2(x, x'),$$

where

$$(z^1(x), z^1(x')) \sim \mathcal{N}\left(0, \begin{bmatrix} \Sigma^1(x, x) & \Sigma^1(x, x') \\ \Sigma^1(x', x) & \Sigma^1(x', x') \end{bmatrix}\right).$$

Now, we assume the results holds for $L$. Then we show the result for $f_\theta^{L+1}(x)$. In this case, we have $f_\theta^{L+1}(x) = g^{L+1}(x)$. By condition on the values $g^L$, we have the output $f_{\theta,k}^{L+1}$ are $i.i.d.$centered Gaussian random variables, $i.e.$,

$$f_{\theta,k}^{L+1}(x) = g_k^{L+1}(x) = \langle w_k^{L+1}, h^L(x) \rangle \overset{i.i.d.}{\sim} \mathcal{N}(0, \|h^L(x)\|^2/n).$$

Then we have

$$\hat{\Sigma}^{L+1}(x, x') = \text{Cov}(f_{\theta,k}^{L+1}(x), f_{\theta,k}^{L+1}(x'))$$
$$= \langle h^L(x), h^L(x') \rangle / n$$
$$= \frac{1}{n} \sum_{i=1}^{n} \phi(g_i^L(x) + g_i^1(x))\phi(g_i^L(x') + g_i^1(x'))$$
$$\overset{a.s.}{\to} \mathbb{E}\phi(z^L(x) + z^1(x))\phi(z^L(x') + z^1(x'))$$
$$=: \Sigma^{L+1}(x, x').$$

where

$$\begin{bmatrix} z^1(x) \\ z^L(x) \\ z^1(x') \\ z^L(x') \end{bmatrix} \sim \mathcal{N}\left(0, \left[\begin{array}{cc|cc} \Sigma^1(x, x) & 0 & \Sigma^1(x, x') & 0 \\ 0 & \Sigma^L(x, x) & 0 & \Sigma^L(x, x') \\ \hline \Sigma^1(x', x) & 0 & \Sigma^1(x', x') & 0 \\ 0 & \Sigma^L(x', x) & 0 & \Sigma^L(x', x') \end{array}\right]\right).$$

Here the covariance is deterministic and independent of $g^L$. Consequently, the conditioned and unconditioned distributions of $f_{\theta,k}^{L+1}$ are equal in the limit: they are $i.i.d.$centered Gaussian random variables with covariance $\Sigma^{L+1}$.

## B.2 Proof of Lemma B.1: the basic case $\ell = 1$

WLOG, we can assume $\sigma_\ell = 1$. We prove by induction. When $\ell = 1$, we have

$$g^1 = W_1 x$$

so that

$$g_k^1 \overset{i.i.d.}{\sim} \mathcal{N}(0, \|x\|^2/n_{in}).$$

Given a controllable function $\Phi$, the random variables $X_k = \Phi(g_k^1)$ are still $i.i.d.$. It follows from Lemma A.6 that

$$\mathbb{E}|X_1| = \mathbb{E}_{z \sim \mathcal{N}(0, \|x\|^2)}|\Phi(z)| \le C_1 e^{C_2^2\|x\|^2} < \infty.$$

Then the desired result is obtained by following Theorem A.3 the classical Kolmogorov's SLLN for $i.i.d.$ random variables.

## B.3 Proof of Lemma B.1: general case for independent matrices $W^k \neq W^\ell$

Suppose the desired result hold for $\ell$, then we show the result also hold for $\ell + 1$. In addition, we assume the weight matrices $W^\ell$ are independent to each other. Thus, the weight matrix $W^{\ell+1}$ are not used in previous layers. For brevity, we denote $W := W^{\ell+1}$ and so we have expression

$$g^{\ell+1} = W h^\ell.$$

Here the randomness of $g^{\ell+1}$ comes from both $W$ and $h^\ell$. As $W$ is not used before, $W$ and $h^\ell$ are independent. Let $\mathcal{B}$ be the $\sigma$-algebra spanned by all previous $g^1, g^2, \cdots, g^\ell$. Then the conditional distribution of $g^{\ell+1}$ on $\mathcal{B}$ is given by

$$g^{\ell+1} | \mathcal{B} \sim \mathcal{N}(0, \|h^\ell\|^2 / n I_n),$$

or equivalently

$$g_k^{\ell+1} | \mathcal{B} \overset{i.i.d.}{\sim} \mathcal{N}(0, \|h^\ell\|^2 / n). \tag{28}$$

By using the inductive hypothesis, we have

$$\sigma_n^2 := \|h^\ell\|^2 / n = \frac{1}{n} \sum_{k=1}^n \phi(g_k^\ell + g_k^1)^2 \overset{a.s.}{\to} \mathbb{E}\left[\phi(z^\ell + z^1)\right]^2 = \Sigma(z^{\ell+1}, z^{\ell+1}) := \sigma^2, \tag{29}$$

where we use the fact $\Phi(x, y) := \phi(x + y)$ is controllable, i.e.,

$$|\Phi(x, y)| = |\phi(x + y)| \leq |x + y| \leq e^{|x| + |y|}.$$

By using triangle inequality, we have

$$\left| \frac{1}{n} \sum_{k=1}^n \Phi(g_k^1, \cdots, g_k^{\ell+1}) - \mathbb{E}\left[\Phi(z^1, \cdots, z^{\ell+1})\right] \right| \leq |A_n| + |B_n| + |C_n|,$$

where

$$A_n = \frac{1}{n} \sum_{k=1}^n \Phi(g_k^1, \cdots, g_k^{\ell+1}) - \frac{1}{n} \sum_{k=1}^n \mathbb{E}_{z \sim \mathcal{N}(0, \sigma_n^2)} \Phi(g_k^1, \cdots, g_k^\ell, z) \tag{30}$$

$$B_n = \frac{1}{n} \sum_{k=1}^n \mathbb{E}_{z \sim \mathcal{N}(0, \sigma_n^2)} \Phi(g_k^1, \cdots, g_k^\ell, z) - \frac{1}{n} \sum_{k=1}^n \mathbb{E}_{z \sim \mathcal{N}(0, \sigma^2)} \Phi(g_k^1, \cdots, g_k^\ell, z) \tag{31}$$

$$C_n = \frac{1}{n} \sum_{k=1}^n \Phi(g_k^1, \cdots, g_k^\ell, z) - \mathbb{E}\left[\Phi(z^1, \cdots, z^{\ell+1})\right] \tag{32}$$

### $A_n$ converges to $0$ almost surely

Define random variables $Z_k := \Phi(g_k^1, \cdots, g_k^\ell, g_k^{\ell+1}) - \mathbb{E}_{z \sim \mathcal{N}(0, \sigma_n^2)} \Phi(g_k^1, \cdots, g_k^\ell, z)$. By equation (28), we have $g_k^{\ell+1} | \mathcal{B} \overset{i.i.d.}{\sim} \mathcal{N}(0, \sigma_n^2)$, we can easily show $X_k$ are centered and uncorrelated. Observe that

$$
\begin{aligned}
\mathbb{E} Z_k &= \mathbb{E}_{\mathcal{B}} \mathbb{E}_{g^{\ell+1} | \mathcal{B}} Z_k \\
&= \mathbb{E}_{\mathcal{B}} \mathbb{E}_{g^{\ell+1} | \mathcal{B}} \left[\Phi(g_k^1, \cdots, g_k^\ell, g_k^{\ell+1}) - \mathbb{E}_{z \sim \mathcal{N}(0, \sigma_n^2)} \Phi(g_k^1, \cdots, g_k^\ell, z)\right] \\
&= \mathbb{E}_{\mathcal{B}} \left[\mathbb{E}_{g^{\ell+1} | \mathcal{B}} \Phi(g_k^1, \cdots, g_k^\ell, g_k^{\ell+1}) - \mathbb{E}_{z \sim \mathcal{N}(0, \sigma_n^2)} \Phi(g_k^1, \cdots, g_k^\ell, z)\right] \\
&= \mathbb{E}_{\mathcal{B}} \left[\mathbb{E}_{z \sim \mathcal{N}(0, \sigma_n^2)} \Phi(g_k^1, \cdots, g_k^\ell, z) - \mathbb{E}_{z \sim \mathcal{N}(0, \sigma_n^2)} \Phi(g_k^1, \cdots, g_k^\ell, z)\right] \\
&= \mathbb{E}_{\mathcal{B}} [0] = 0.
\end{aligned}
$$

Similarly, we obtain $\mathbb{E}Z_k Z_{k'} = \delta_{kk'} \mathbb{E} |Z_k|^2$. Moreover, we can upper bound $\mathbb{E}[Z_k|\mathcal{B}]^2$ as follows

$$
\begin{aligned}
\mathbb{E}[Z_k|\mathcal{B}]^2 &= \mathbb{E}_{g^{\ell+1}|\mathcal{B}} \left| \Phi(g_k^1, \cdots, g_k^\ell, g_k^{\ell+1}) - \mathbb{E}_{z \sim \mathcal{N}(0,\sigma_n^2)} \Phi(g_k^1, \cdots, g_k^\ell, z) \right|^2 \\
&\leq 8\mathbb{E}_{z \sim \mathcal{N}(0,\sigma_n^2)} \left| \Phi(g_k^1, \cdots, g_k^\ell, z) \right|^2, \quad (a) \\
&= 8\mathbb{E}_{z \sim \mathcal{N}(0,1)} \left| \Phi(g_k^1, \cdots, g_k^\ell, \sigma_n z) \right|^2 \\
&\leq 8\mathbb{E}_{z \sim \mathcal{N}(0,1)} C_1 e^{2C_2(\sum_{i=1}^\ell |g_k^i| + \sigma_n |z|)}, \quad \Phi \text{ is controllable} \\
&= 8C_1 e^{2C_2 \sum_{i=1}^\ell |g_k^i|} \mathbb{E}_{z \sim \mathcal{N}(0,1)} e^{2C_2 \sigma_n |z|} \\
&\leq 8C_1 e^{2C_2 \sum_{i=1}^\ell |g_k^i|} e^{2C_2^2 \sigma_n^2}.
\end{aligned}
$$

where $(a)$ is due to maximal and Jensen's inequality.

Since $e^{2C_2 \sum_{i=1}^\ell |g_k^i|}$ is controllable and $\sigma_n \overset{a.s.}{\to} \sigma$, it follows from the inductive hypothesis that

$$
\frac{1}{n} \sum_{k=1}^n \mathbb{E}[Z_k|\mathcal{B}]^2 \leq 8C_1 \cdot \left( \frac{1}{n} \sum_{k=1}^n e^{2C_2 \sum_{i=1}^\ell |g_k^i|} \right) \cdot e^{2C_2^2 \sigma_n^2} \overset{a.s.}{\to} 8C_1 \mathbb{E}e^{2C_2 \sum_{i=1}^\ell |z_i|} \cdot e^{2C_2^2 \sigma^2}.
$$

As the RHS is a deterministic constant, we have

$$
\frac{1}{n} \sum_{k=1}^n \mathbb{E}[Z_k|\mathcal{B}]^2 \in o(n^\rho), \quad \forall \rho > 0.
$$

or equivalently, $\frac{1}{n} \sum_{k=1}^n \mathbb{E}[Z_k|\mathcal{B}]^2 \leq n^\rho$ for large enough $n$.

Now, we will first show $A_{n^2} \overset{a.s.}{\to} 0$. For any $\epsilon > 0$, we have for large enough $n$

$$
\begin{aligned}
\mathbb{P}(|A_{n^2}| \geq \epsilon) &\leq \epsilon^{-2} n^{-4} \mathbb{E}|A_{n^2}|^2 \\
&= \epsilon^{-2} n^{-4} \sum_{k,k'=1}^{n^2} \mathbb{E}[Z_k Z_{k'}] \\
&= \epsilon^{-2} n^{-4} \sum_{k=1}^{n^2} \mathbb{E}|Z_k|^2 \\
&= \epsilon^{-2} n^{-2} \mathbb{E}_{\mathcal{B}} \left[ \frac{1}{n^2} \sum_{k=1}^{n^2} \mathbb{E}|Z_k|\mathcal{B}|^2 \right] \\
&= \epsilon^{-2} n^{-2} \mathbb{E}_{\mathcal{B}} \left[ n^{2\rho} \right] \\
&\leq \epsilon^{-2} n^{-2+2\rho}.
\end{aligned}
$$

Furthermore, we obtain

$$
\sum_{n=1}^\infty \mathbb{P}(|A_{n^2}| \geq \epsilon) \leq \sum_{n=1}^\infty \epsilon^{-2} n^{-2+2\rho} < \infty,
$$

provided we choose $0 < \rho < 1/2$. Thus, it follows from Borel-Cantelli lemma that $A_{n^2} \overset{a.s.}{\to} 0$.

Now for each $n$, we define $k_n := \sup\{k \in \mathbb{N} : k^2 \leq n\}$, then we have $k_n^2 \leq n \leq (k_n + 1)^2$. Note that

$$
A_n = \frac{1}{n} \sum_{i=1}^n Z_i = \frac{1}{n} \sum_{i=1}^{k_n^2} Z_i + \frac{1}{n} \sum_{i=k_n^2+1}^n Z_i.
$$

We will show the two terms goes 0 a.s.. As we just proved, the first term goes to 0 a.s., since

$$
\left| \frac{1}{n} \sum_{i=1}^{k_n^2} Z_i \right| \leq \left| \frac{1}{k_n^2} \sum_{i=1}^{k_n^2} Z_i \right| \overset{a.s.}{\to} 0.
$$

For the second term, let $T_n := \frac{1}{n} \sum_{i=k_n^2+1}^{n} Z_i$, then for $n$ large enough

$$\mathbb{P}(|T_n| \geq \epsilon) \leq \epsilon^{-2} n^{-2} \sum_{i=k_n^2+1}^{n} \mathbb{E} Z_i^2$$

$$\leq \epsilon^{-2} k_n^{-4} \sum_{i=k_n^2+1}^{n} \mathbb{E} Z_i^2$$

$$\leq \epsilon^{-2} k_n^{-4} \left(n - k_n^2\right) \left(\frac{1}{n - k_n^2} \sum_{i=k_n^2+1}^{n} \mathbb{E} Z_i^2\right)$$

$$\leq \epsilon^{-2} k_n^{-4} \left(n - k_n^2\right)^{1+\rho}$$

$$\leq C\epsilon^{-2} k_n^{-4} \left(2k_n + 1\right)^{1+\rho}$$

$$\leq C\epsilon^{-2} k_n^{-3+\rho}$$

where $C$ is some fixed constant. Then we have

$$\sum_{n=1}^{\infty} \mathbb{P}(|T_n| \geq \epsilon) \leq \sum_{n=1}^{\infty} C\epsilon^{-2} k_n^{-3+\rho}$$

$$\leq \sum_{n=1}^{\infty} C\epsilon^{-2}(\sqrt{n} - 1)^{-3+\rho}$$

$$\leq \sum_{n=1}^{4} C\epsilon^{-2}(\sqrt{n} - 1)^{-3+\rho} + 2C\epsilon^{-2} \sum_{n=4}^{\infty} n^{-(3-\rho)/2}$$

$$< \infty,$$

provided we choose $0 < \rho < 1$. Therefore, by choosing $0 < \rho < 1/2$, it follows from Borel-Cantelli lemma that $T_n \overset{a.s.}{\to} 0$ and further $A_n \overset{a.s.}{\to} 0$.

## $B_n$ **converges to** $0$ **almost surely**

First of all, we will show $\sigma > 0$ by which we can use Gaussian smoothing to show $B_n \overset{a.s.}{\to} 0$.

**Lemma B.3.** *For $\ell \geq 1$, if $\Sigma(z^\ell, z^\ell) > 0$, then $\Sigma(z^{\ell+1}, z^{\ell+1}) > 0$.*

*Proof.* We prove by contradiction. Assume $\Sigma(z^{\ell+1}, z^{\ell+1}) = 0$. Then we have

$$0 = \Sigma(z^{\ell+1}, z^{\ell+1}) = \mathbb{E}\phi(z^\ell + z^1)^2 = \mathbb{E}\phi(u^\ell)^2,$$

where $u^\ell \sim \mathcal{N}(0, \Sigma(z^\ell, z^\ell) + \|x\|^2/n_{in})$. It implies $\phi(z) = 0$ almost surely, but it contradicts $\phi$ is non-constant function since $\Sigma(z^\ell, z^\ell) + \|x\|^2/n_{in} > 0$. $\qquad\square$

It follows from Lemma B.3 that $\sigma > 0$. Then $\sigma_n \overset{a.s.}{\to} \sigma$, we have $\sigma_n \geq \sigma/2 > 0$ eventually, almost surely. To use Gaussian smoothing, we define following functions

$$f_k(x) := \Phi(g_k^1, \cdots, g_k^\ell, x), \quad F_k(\sigma) := \mathbb{E}_{z\sim\mathcal{N}(0,\sigma^2)} f_k(z).$$

By using Gaussian smoothing, we have for large enough $n$

$$|B_n| \leq \frac{1}{n} \sum_{k=1}^{n} |F_k(\sigma_n) - F_k(\sigma)|$$

$$\leq \frac{1}{n} \sum_{k=1}^{n} \int_{\sigma}^{\sigma_n} |F_k'(t)| \, dt, \quad \text{assume } \sigma \leq \sigma_n$$

$$\leq \frac{1}{n} \sum_{k=1}^{n} \int_{\sigma}^{\sigma_n} t^{-1} \mathbb{E}_{z \sim \mathcal{N}(0,1)} |f_k(tz)(t^2 - 1)| \, dt, \quad (a)$$

$$\leq \frac{1}{n} \sum_{k=1}^{n} \int_{\sigma}^{\sigma_n} t^{-1} \mathbb{E}_{z \sim \mathcal{N}(0,1)} C_1 e^{C_2 \sum_{i=1}^{\ell} |g_k^i| + C_2 t |z| + t} \, dt, \quad (b)$$

$$\leq \frac{1}{n} \sum_{k=1}^{n} \int_{\sigma}^{\sigma_n} t^{-1} C_1 e^{C_2 \sum_{i=1}^{\ell} |g_k^i| + C_2 t^2/2 + t} \, dt, \quad (c)$$

$$= C_1 \left( \frac{1}{n} \sum_{k=1}^{n} e^{C_2 \sum_{i=1}^{\ell} |g_k^i|} \right) (\alpha(\sigma_n) - \alpha(\sigma))$$

where $(a)$ is by Lemma A.2 and Jensen's inequality, $(b)$ is because $f_k$ is controllable since $\Phi$ is, $(c)$ is by Lemma A.6, and $\alpha(t)$ is the anti-derivative of the function $\dot{\alpha}(t) = t^{-1} C_1 e^{C_2 t^2/2 + t}$. Here, $\dot{\alpha}(t)$ is continuous, so that $\alpha(t)$ is well-defined and continuous. Since $e^{C_2 \sum_{i=1}^{\ell} |g_k^i|}$ is controllable, it follows from result for the basic case that

$$\frac{1}{n} \sum_{k=1}^{n} e^{C_2 \sum_{i=1}^{\ell} |g_k^i|} \xrightarrow{a.s.} \mathbb{E}_{z \sim \mathcal{N}(0, \Sigma|g^1)} e^{C_2 \sum_{i=1}^{\ell} |z^i|}.$$

Since $\sigma_n \xrightarrow{a.s.} \sigma$ and $\alpha$ is continuous, it follows from Lemma A.1 that $\alpha(\sigma_n) \xrightarrow{a.s.} \alpha(\sigma)$ and further

$$|B_n| \leq C_1 \left( \frac{1}{n} \sum_{k=1}^{n} e^{C_2 \sum_{i=1}^{\ell} |g_k^i|} \right) (\alpha(\sigma_n) - \alpha(\sigma)) \xrightarrow{a.s.} 0.$$

### $C_n$ converges to $0$ almost surely

Define function $\hat{\Phi}(z^1, \cdots, z^\ell) := \mathbb{E}_{z \sim \mathcal{N}(0,1)} \Phi(z^1, \cdots, z^\ell, \sigma z)$. Since $\Phi$ is controllable, $\hat{\Phi}$ is also a controllable function. Then it follows from the inductive hypothesis that

$$\frac{1}{n} \sum_{k=1}^{n} \mathbb{E}_{z \sim \mathcal{N}(0, \sigma^2)} \Phi(g_k^1, \cdots, g_k^\ell, z) = \frac{1}{n} \sum_{k=1}^{n} \mathbb{E}_{z \sim \mathcal{N}(0,1)} \Phi(g_k^1, \cdots, g_k^\ell, \sigma z)$$

$$= \frac{1}{n} \sum_{k=1}^{n} \hat{\Phi}(g_k^1, \cdots, g_k^\ell)$$

$$\xrightarrow{a.s.} \mathbb{E} \left[ \hat{\Phi}(z^1, \ldots, z^\ell) \right]$$

$$= \mathbb{E} \left[ \mathbb{E}_{z \sim \mathcal{N}(0,1)} \Phi(z^1, \cdots, z^\ell, \sigma z) \right]$$

$$= \mathbb{E} \left[ \Phi(z^1, \cdots, z^\ell, z^{\ell+1}) \right]$$

Thus, $C_n \xrightarrow{a.s.} 0$.

### B.4   Proof of Lemma B.1: general case for shared matrices

Now in this section, we prove the desired result when the weight matrices are shared, *i.e.*, $W^\ell = W$. Assume the result holds for $\ell$, then we will show the desired result still holds for $\ell + 1$. Note that

$$g^{\ell+1} = W h^\ell.$$

As $W$ is used before, we have
$$g^i = Wh^{i-1}, \quad \forall i \in [\ell].$$
Then define
$$G := \begin{bmatrix} g^1 & g^2 & \cdots & g^\ell \end{bmatrix} \in \mathbb{R}^{n \times \ell}, \quad H := \begin{bmatrix} h^0 & h^1 & \cdots & h^{\ell-1} \end{bmatrix} \in \mathbb{R}^{n \times \ell}. \tag{33}$$
Then we have $G = WH$. Let $\mathcal{B}$ be the $\sigma$-algebra spanned by all previous $g^1, g^2, \cdots, g^\ell$. To obtain the conditional distribution of $g^{\ell+1}$ on $\mathcal{B}$, we first compute the conditional distribution of $W$ on $\mathcal{B}$. It follows from Lemma A.3 that
$$\begin{aligned} W | \mathcal{B} &= G \left( H^T H \right)^\dagger H^T + \tilde{W} \Pi_H^T \\ &\sim \mathcal{MN} \left( G(H^T H)^\dagger H^T, I_n, \Pi_H \Pi_H^T / n \right) \end{aligned}$$
where $\Pi = I_n - HH^\dagger$ is the orthogonal projection onto null$(H^T)$, respectively. Therefore, we obtain
$$g^{\ell+1} | \mathcal{B} \sim \mathcal{N} \left( G \left( H^T H \right)^\dagger H^T h^\ell, \|\Pi^T h^\ell\|^2 / n I_n \right)$$
or equivalently
$$g_k^{\ell+1} | \mathcal{B} \overset{\text{independent}}{\sim} \mathcal{N} \left( G_k \left( H^T H \right)^\dagger H^T h^\ell, \|\Pi^T h^\ell\|^2 / n \right),$$
where $G_k \in \mathbb{R}^{1 \times \ell}$ is the $k$-th row of $G$.

Since the activation function $\phi$ is controllable by Lemma A.5, it follows from the inductive hypothesis that
$$(h^i)^T (h^j)/n = \frac{1}{n} \sum_{k=1}^n \phi(g_k^i + g_k^1) \phi(g_k^j + g_k^1) \xrightarrow{a.s.} \mathbb{E}\phi(z^i + z^1)\phi(z^j + z^1) = \Sigma(z^{i+1}, z^{j+1}) \quad \forall i, j.$$
Then we have as $n \to \infty$
$$H^T H / n \xrightarrow{a.s.} \Sigma(Z^\ell, Z^\ell)$$
$$H^T h^\ell / n \xrightarrow{a.s.} \Sigma(Z^\ell, z^{\ell+1})$$
where $Z^\ell = [z^1 \cdots z^\ell]^T \in \mathbb{R}^{\ell \times 1}$. Since (pseudo-)inverse is continuous function, we further obtain
$$v_n := \left( H^T H \right)^\dagger H^T h^\ell = \left( H^T H/n \right)^\dagger H^T h^\ell / n \xrightarrow{a.s.} \Sigma(Z^\ell, Z^\ell)^\dagger \Sigma(Z^\ell, z^{\ell+1}) := v. \tag{34}$$
By using the equality $HH^\dagger = H(H^T H)^\dagger H^T$, we have
$$\begin{aligned} \|\Pi^T h^\ell\|^2 / n &= \frac{1}{n} (h^\ell)^T \left( I_n - HH^\dagger \right)^2 h^\ell \\ &= \frac{1}{n} (h^\ell)^T \left( I_n - HH^\dagger \right) h^\ell \\ &= \frac{1}{n} (h^\ell)^T h^\ell - \left( (h^\ell)^T H/n \right) (H^T H/n)^\dagger \left( H^T h^\ell / n \right) \\ &\xrightarrow{a.s.} \Sigma(z^{\ell+1}, z^{\ell+1}) - \Sigma(z^{\ell+1}, Z^\ell)\Sigma(Z^\ell, Z^\ell)^\dagger \Sigma(Z^\ell, z^{\ell+1}). \end{aligned}$$

By using triangular inequality, we have
$$\left| \frac{1}{n} \sum_{k=1}^n \Phi(g_k^1, \cdots, g_k^\ell, g_k^{\ell+1}) - \mathbb{E}\left[ \Phi(z^1, \cdots, z^{\ell+1}) \right] \right| \leq |A_n| + |B_n| + |C_n| + |D_n|,$$
where
$$A_n = \frac{1}{n} \sum_{k=1}^n \Phi(g_k^1, \cdots, g_k^\ell, g_k^{\ell+1}) - \frac{1}{n} \sum_{k=1}^n \mathbb{E}_{z \sim \mathcal{N}(\mu_{k,n}, \sigma_n^2)} \Phi(g_k^1, \cdots, g_k^\ell, z) \tag{35}$$
$$B_n = \frac{1}{n} \sum_{k=1}^n \mathbb{E}_{z \sim \mathcal{N}(\mu_{k,n}, \sigma_n^2)} \Phi(g_k^1, \cdots, g_k^\ell, z) - \frac{1}{n} \sum_{k=1}^n \mathbb{E}_{z \sim \mathcal{N}(\mu_{k,n}, \sigma^2)} \Phi(g_k^1, \cdots, g_k^\ell, z) \tag{36}$$
$$C_n = \frac{1}{n} \sum_{k=1}^n \mathbb{E}_{z \sim \mathcal{N}(\mu_{k,n}, \sigma^2)} \Phi(g_k^1, \cdots, g_k^\ell, z) - \frac{1}{n} \sum_{k=1}^n \mathbb{E}_{z \sim \mathcal{N}(\mu_k, \sigma^2)} \Phi(g_k^1, \cdots, g_k^\ell, z) \tag{37}$$
$$D_n = \frac{1}{n} \sum_{k=1}^n \mathbb{E}_{z \sim \mathcal{N}(\mu_k, \sigma^2)} \Phi(g_k^1, \cdots, g_k^\ell, z) - \mathbb{E}\left[ \Phi(z^1, \cdots, z^{\ell+1}) \right] \tag{38}$$

where

$$\mu_{k,n} = G_k^\ell (H^T H)^\dagger H^T h_\ell = G_k^\ell v_n, \tag{39}$$

$$\mu_k = G_k^\ell \Sigma(Z^\ell, Z^\ell)^\dagger \Sigma(Z^\ell, z^{\ell+1}) = G_k^\ell v, \tag{40}$$

$$\sigma_n^2 = \|\Pi^T h^\ell\|^2 \tag{41}$$

$$\sigma^2 = \Sigma(z^{\ell+1}, z^{\ell+1}) - \Sigma(z^{\ell+1}, Z^\ell)\Sigma(Z^\ell, Z^\ell)^\dagger \Sigma(Z^\ell, z^{\ell+1}). \tag{42}$$

### B.4.1 $A_n$ converges to $0$ almost surely

Define random variables $Z_k = \Phi(g_k^1, \cdots, g_k^\ell, g_k^{\ell+1}) - \mathbb{E}_{z\sim\mathcal{N}(\mu_{k,n},\sigma_n^2)}\Phi(g_k^1, \cdots, g_k^\ell, z)$. As $X_k|\mathcal{B}$ are independent, we can easily show $X_k$ are centered and uncorrelated. By using Jensen's inequality, $Z_k^2|\mathcal{B}$ can be upper bounded as follows

$$\mathbb{E}\left[Z_k^2|\mathcal{B}\right] \leq 8\mathbb{E}_{z\sim\mathcal{N}(\mu_{k,n},\sigma_n^2)}\left|\Phi(g_k^1, \cdots, g_k^\ell, z)\right|^2 \leq 8C_1 e^{2C_2 \sum_{i=1}^\ell |g_k^i|} e^{2C_2|\mu_{k,n}|} e^{2C_2^2 \sigma_n^2} \tag{43}$$

As $v_n \overset{a.s.}{\to} v$ by equation (34), we have $\|v_n\| \leq 1 + \|v\|$, eventually, almost surely. Thus, for large enough $n$, we have

$$|\mu_{k,n}| = \left|G_k^\ell(H^T H)^\dagger(H^T h^\ell)\right| = \left|\sum_{i=1}^\ell v_{n,i} g_k^i\right| \leq (\|v\| + 1)\sum_{i=1}^\ell |g_k^i|, \tag{44}$$

where we also use the Cauchy-Schwartz inequality and square root inequality. It follows from equation (43) that

$$\mathbb{E}\left[Z_k^2|\mathcal{B}\right] \leq 8C_1 e^{(2C_2+\|v\|+1)\sum_{i=1}^\ell |g_k^i|} e^{2C_2^2 \sigma_n^2} = \hat{\Phi}(g_k^1, \cdots, g_k^\ell) \cdot e^{2C_2^2 \sigma_n^2},$$

where $\hat{\Phi}(x^1, \cdots, x^\ell) := 8C_1 e^{(2C_2+\|v\|+1)\sum_{i=1}^\ell |x_i|}$ is clearly a controllable function. It follows from inductive hypothesis and some basic properties of almost surely convergence in Lemma A.1 that

$$\frac{1}{n}\sum_{k=1}^n \mathbb{E}\left[Z_k^2|\mathcal{B}\right] \leq \frac{1}{n}\sum_{k=1}^n \hat{\Phi}(g_k^1, \cdots, g_k^\ell) \cdot e^{2C_2^2 \sigma_n^2} \overset{a.s.}{\to} \mathbb{E}\left[\hat{\Phi}(z^1, \cdots, z^\ell)\right] \cdot e^{2C_2^2 \sigma^2}.$$

As RHS is a deterministic constant, we have $\frac{1}{n}\sum_{k=1}^n \mathbb{E}\left[X_k^2|\mathcal{B}\right] \in o(n^\rho)$ for all $\rho > 0$. Then by using the same argument provided in Section B.3, we have $A_n \overset{a.s.}{\to} 0$.

### $B_n$ converges to $0$ almost surely

**If $\sigma > 0$**

In this subsection, we assume $\sigma > 0$. In addition, since $\sigma_n \overset{a.s.}{\to} \sigma$, we have $\sigma_n \geq \sigma/2 > 0$ almost surely for large enough $n$.

We can obtain the desired result $B_n \overset{a.s.}{\to} 0$ by applying the same argument in Section B.3 to functions $f_k$ and $F_k$ redefined as follows

$$f_k(x) := \Phi(g_k^1, \cdots, g_k^\ell, x), \quad F_k(\sigma) := \mathbb{E}_{z\sim\mathcal{N}(\mu_{k,n},\sigma^2)} f_k(z).$$

By using Gaussian smoothing, for large enough $n$, we have

$$
\begin{aligned}
|B_n| &\leq \frac{1}{n} \sum_{k=1}^{n} |F_k(\sigma_n) - F_k(\sigma)| \\
&\leq \frac{1}{n} \sum_{k=1}^{n} \int_{\sigma}^{\sigma_n} |F_k'(t)| \, dt, \quad \text{assume } \sigma \leq \sigma_n \\
&\leq \frac{1}{n} \sum_{k=1}^{n} \int_{\sigma}^{\sigma_n} t^{-1} \mathbb{E}_{z \sim \mathcal{N}(0,1)} \left| f_k(\mu_{k,n} + tz)(t^2 - 1) \right| dt, \quad (a) \\
&\leq \frac{1}{n} \sum_{k=1}^{n} \int_{\sigma}^{\sigma_n} t^{-1} \mathbb{E}_{z \sim \mathcal{N}(0,1)} C_1 e^{(C_2 + \|v\| + 1) \sum_{i=1}^{\ell} |g_k^i| + C_2 t |z| + t} dt, \quad (b) \\
&\leq \frac{1}{n} \sum_{k=1}^{n} \int_{\sigma}^{\sigma_n} t^{-1} C_1 e^{(C_2 + \|v\| + 1) \sum_{i=1}^{\ell} |g_k^i| + C_2 t^2/2 + t} dt, \quad (c) \\
&= C_1 \left( \frac{1}{n} \sum_{k=1}^{n} e^{(C_2 + \|v\| + 1) \sum_{i=1}^{\ell} |g_k^i|} \right) (\alpha(\sigma_n) - \alpha(\sigma)),
\end{aligned}
$$

where $(a)$ is by Lemma A.2, $(b)$ is because $f_k$ is controllable since $\Phi$ is, $(c)$ is by Lemma A.6 and equation (44), and $\alpha(t)$ is the anti-derivative of the function $\dot{\alpha}(t) = t^{-1} C_1 e^{C_2 t^2/2 + t}$. Here, $\dot{\alpha}(t)$ is continuous, so that $\alpha(t)$ is well-defined and continuous. Since $e^{C \sum_{i=1}^{\ell} |g_k^i|}$ is controllable for any constant $C$, it follows from the inductive hypothesis that

$$
\frac{1}{n} \sum_{k=1}^{n} e^{(C_2 + \|v\| + 1) \sum_{i=1}^{\ell} |g_k^i|} \xrightarrow{a.s.} \mathbb{E} \left[ e^{(C_2 + \|v\| + 1) \sum_{i=1}^{\ell} |z_i|} \right] < \infty.
$$

Since $\sigma_n \xrightarrow{a.s.} \sigma$ and $\alpha$ is continuous, it follows from Lemma A.1 that $\alpha(\sigma_n) \xrightarrow{a.s.} \alpha(\sigma)$ and further

$$
|B_n| \leq C_1 \left( \frac{1}{n} \sum_{k=1}^{n} e^{(C_2 + \|v\| + 1) \sum_{i=1}^{\ell} |g_k^i|} \right) (\alpha(\sigma_n) - \alpha(\sigma)) \xrightarrow{a.s.} 0.
$$

**If $\sigma = 0$**

In this subsection, we consider when $\sigma = 0$. Note that the argument in the case $\sigma > 0$ also holds if $\sigma = 0$ and $\sigma_n \neq 0$ (infinitely often), because the derivatives $F_k'(t)$ are well-defined if either $\sigma > 0$ or $\sigma_n > 0$. Thus, we only need to analyze the case where $\sigma = 0$ and $\sigma_n = 0$ eventually.

For $\sigma = 0$, we have $\Sigma(z^{\ell+1}, z^{\ell+1}) = \Sigma(z^{\ell+1}, Z^\ell) \Sigma(Z^\ell, Z^\ell)^\dagger \Sigma(Z^\ell, z^{\ell+1})$. By Lemma A.4, we have

$$
z^{\ell+1} = \Sigma(z^{\ell+1}, Z^\ell) \Sigma(Z^\ell, Z^\ell)^\dagger Z^\ell = v Z^\ell, \quad a.s.
$$

For controllable $\Phi$, we can show the function $\hat{\Phi} : (g_k^1, \cdots, g_k^\ell) \mapsto \Phi(g_k^1, \cdots, g_k^\ell, G_k^\ell v_n)$ is also controllable as follows

$$
\begin{aligned}
\left| \hat{\Phi}(g_k^1, \cdots, g_k^\ell) \right| &= \left| \Phi(g_k^1, \cdots, g_k^\ell, G_k^\ell v_n) \right| \\
&\leq C_1 e^{C_2 \sum_{i=1}^{\ell} |g_k^i| + C_2 |\sum_{i=1}^{\ell} v_{n,i} g_k^i|} \\
&\leq C_1 e^{(2C_2 + \|v\| + 1) \sum_{i=1}^{\ell} |g_k^i|},
\end{aligned}
$$

where the second inequality follows from equation (34). By using the inductive hypothesis, we obtain

$$
\begin{aligned}
\frac{1}{n} \sum_{k=1}^{n} \Phi(g_k^1, \cdots, g_k^\ell, G_k^\ell v_n) &= \frac{1}{n} \sum_{k=1}^{n} \hat{\Phi}(g_k^1, \cdots, g_k^\ell) \\
&\xrightarrow{a.s.} \mathbb{E} \left[ \hat{\Phi}(z^1, \cdots, z^\ell) \right] \\
&= \mathbb{E} \left[ \Phi(z^1, \cdots, z^\ell, v Z^\ell) \right] \\
&= \mathbb{E} \left[ \Phi(z^1, \cdots, z^{\ell+1}) \right].
\end{aligned} \tag{45}
$$

Moreover, as we assume $\sigma_n = 0$ for all large enough $n$, we obtain $g_k^{\ell+1}|\mathcal{B} = G_k^\ell v_n$ almost surely. Then for large enough $n$, we obtain

$$\frac{1}{n}\sum_{k=1}^{n}\mathbb{E}_{z\sim\mathcal{N}(\mu_{k,n},\sigma_n^2)}\Phi(G_k^\ell,z) = \frac{1}{n}\sum_{k=1}^{n}\Phi(g_k^1,\cdots,g_k^\ell,\mu_{k,n}) = \frac{1}{n}\sum_{k=1}^{n}\Phi(g_k^1,\cdots,g_k^\ell,G_k^\ell v_n) \quad (46)$$

Combining $A_n \overset{a.s.}{\to} 0$ with equations (45) and (46) yields $B_n \overset{a.s.}{\to} 0$.

### B.4.2 $\quad C_n$ converges to $0$ almost surely

As discussed in Section B.4.1, we can assume $\sigma > 0$. By using Gaussian smoothing again, we can easiliy show $C_n \overset{a.s.}{\to} 0$ since $\mu_{k,n} \overset{a.s.}{\to} \mu_k$. Define functions

$$f_k(x) = \Phi(g_k^1,\cdots,g_k^\ell,x), \quad F_k(\mu) = \mathbb{E}_{z\sim\mathcal{N}(\mu,\sigma^2)}f_k(z).$$

It follows from Lemma A.2 that

$$
\begin{aligned}
|C_n| &\leq \frac{1}{n}\sum_{k=1}^{n}|F_k(\mu_{k,n}) - F_k(\mu)| \\
&\leq \frac{1}{n}\sum_{k=1}^{n}\int_{\mu_k}^{\mu_{k,n}}|F_k'(t)|\,dt, \quad \text{assume } \mu_k \leq \mu_{k,n} \\
&\leq \frac{1}{n}\sum_{k=1}^{n}\int_{\mu_k}^{\mu_{k,n}}\frac{1}{\sigma}\mathbb{E}_{z\sim\mathcal{N}(0,1)}|f_k(t+\sigma z)|\,|z|\,dt \\
&\leq \frac{1}{n}\sum_{k=1}^{n}\int_{\mu_k}^{\mu_{k,n}}\frac{1}{\sigma}\mathbb{E}_{z\sim\mathcal{N}(0,1)}C_1 e^{C_2\sum_{i=1}^{\ell}|g_k^i|+C_2 t+(C_2\sigma+1)|z|}\,dt \\
&\leq \frac{1}{\sigma}C_1 e^{(C_2\sigma+1)^2/2} \cdot \frac{1}{n}\sum_{k=1}^{n}e^{C_2\sum_{i=1}^{\ell}|g_k^i|} \cdot [\beta(\mu_{k,n}) - \beta(\mu_k)],
\end{aligned}
$$

where $\beta(\mu)$ is the anti-derivative of the function $\dot\beta(t) = e^{C_2 t}$. Here $\beta$ is well-defined and continuous since $\dot\beta$ is continuous. As $\mu_{k,n} \overset{a.s.}{\to} \mu_k$, it follows from inductive hypothesis and Lemma A.1 that $C_n \overset{a.s.}{\to} 0$.

### $D_n$ converges to $0$ almost surely

In this section, we can show $D_n \overset{a.s.}{\to} 0$ straightforward from the induction. Define functions

$$\hat\Phi(z^1,\cdots,z^\ell) := \mathbb{E}_{z\sim\mathcal{N}(0,1)}\left[\Phi\left(z^1,\cdots,z^\ell,\sum_{i=1}^{\ell}v_i z_i + \sigma z\right)\right].$$

Here $\hat\Phi$ is controllable as $\Phi$ is. By applying the inductive hypothesis on $\hat\Phi$, we obtain

$$
\begin{aligned}
D_n &= \left|\frac{1}{n}\sum_{k=1}^{n}\mathbb{E}_{z\sim\mathcal{N}(\mu_k,\sigma^2)}\Phi(g_k^1,\cdots,g_k^\ell,z) - \mathbb{E}\left[\Phi(z^1,\cdots,z^{\ell+1})\right]\right| \\
&= \left|\frac{1}{n}\sum_{k=1}^{n}\mathbb{E}_{z\sim\mathcal{N}(0,1)}\Phi(g_k^1,\cdots,g_k^\ell,\mu_k+\sigma z) - \mathbb{E}_{z^1,\cdots,z^\ell}\mathbb{E}_{z^{\ell+1}|z^1,\cdots,z^\ell}\Phi(z^1,\cdots,z^{\ell+1})\right| \\
&= \left|\frac{1}{n}\sum_{k=1}^{n}\mathbb{E}_{z\sim\mathcal{N}(0,1)}\Phi(g_k^1,\cdots,g_k^\ell,\mu_k+\sigma z) - \mathbb{E}_{z^1,\cdots,z^\ell}\mathbb{E}_{z\sim\mathcal{N}(0,1)}\Phi(z^1,\cdots,\sigma z)\right| \\
&= \left|\frac{1}{n}\sum_{k=1}^{n}\hat\Phi(g_k^1,\cdots,g_k^\ell) - \mathbb{E}_{z^1,\cdots,z^\ell}\hat\Phi(z^1,\cdots,z^\ell)\right| \\
&\overset{a.s.}{\to} 0,
\end{aligned}
$$

where we use the fact $\mu_k = G_k^\ell v = \sum_{i=1}^{\ell}v_i g_k^i$.

# C  Proof of Corollary 4.2

Define Gaussian random variables $u^\ell(x)$ that is encoded by input $x$ as follows for all $\ell = [2, L-1]$

$$u^1(x) = z^1(x) \tag{47}$$

$$u^\ell(x) = z^\ell(x) + z^1(x). \tag{48}$$

Then we can easily compute the corresponding covariance as follows for $\ell \geq 2$

$$\begin{aligned}
\mathrm{cov}(u^1(x), u^1(x')) &= \mathrm{cov}(z^1(x), z^1(x')) \\
&= \Sigma^1(x, x') \\
\mathrm{cov}(u^\ell(x), u^\ell(x')) &= \mathrm{cov}(z^\ell(x) + z^1(x), z^\ell(x') + z^1(x')) \\
&= \mathrm{cov}(z^\ell(x), z^\ell(x')) + \mathrm{cov}(z^1(x), z^1(x')) \\
&= \Sigma^\ell(x, x') + \Sigma^1(x, x')
\end{aligned}$$

# D  Proof of Theorem 4.3

This section is deducted to prove the strict positive definiteness of $\Sigma^L$. We will prove it by using the notion of *dual activation* and *Hermitian expansion*.

Let $x \sim \mathcal{N}(0, 1)$ and $f : \mathbb{R} \to \mathbb{R}$. Then we can define an inner product

$$\langle f, g \rangle := \mathbb{E}_{x \sim \mathcal{N}(0,1)} f(x) g(x).$$

Thus, we define a Hilbert space of functions $\mathcal{H}$, that is, $f \in \mathcal{H}$ if and only if

$$\|f\|^2 = \mathbb{E}_{x \sim \mathcal{N}(0,1)} |f(x)|^2 < \infty.$$

Next, consider the function sequence $1, x, x^2, \cdots$. Clearly, they are independent. Then apply Gram-Schmidt process to the function sequence w.r.t. the inner product we defined before, and we obtain $\{h_n\}$ the **(normalized) Hermite polynomial** that is an **orthonormal basis** to the Hilbert space $\mathcal{H}$.

Now, we are ready to introduce *dual activation*. The **dual activation** $\hat{\phi} : [-1, 1] \to \mathbb{R}$ of an activation $\phi : \mathbb{R} \to \mathbb{R}$ is defined by

$$\hat{\phi}(\rho) := \mathbb{E}_{(X,Y) \sim \mathcal{N}_\rho} \phi(X)\phi(Y). \tag{49}$$

where $\mathcal{N}_\rho$ is multidimensional Gaussian distribution with mean 0 and covariance matrix $\begin{bmatrix} 1 & \rho \\ \rho & 1 \end{bmatrix}$.

Then the **dual kernel** $k_\phi$ is given by

$$k_\phi(x, x') := \hat{\phi}(\langle x, x' \rangle).$$

If a function $\phi \in \mathcal{H}$, we not only can obtain an expansion by using the orthonormal basis of Hermitian polynomials but also an expansion to the dual activation $\hat{\phi}$ by using the same Hermitian coefficients. As a consequence, the corresponding dual kernel $k_\phi$ can be shown to be strict positive definite by using the Hermitian expansion.

**Lemma D.1.** *[10, Lemma 12] If $\phi \in \mathcal{H}$, then*

$$\phi(x) = \sum_{n=0}^{\infty} a_n h_n(x), \tag{50}$$

$$\hat{\phi}(\rho) = \sum_{n=0}^{\infty} a_n^2 \rho^n. \tag{51}$$

*where $a_n := \langle h_n, \phi \rangle$ is the **Hermite coefficients**, and the above is **Hermitian expansion**.*

**Theorem D.1.** *[23, Theorem 3][17, Theorem 1] For a function $f : [-1, 1] \to \mathbb{R}$ with $f = \sum_{n=0}^{\infty} b_n h_n$, the kernel $K_f : S^{n_0-1} \times S^{n_0-1} \to \mathbb{R}$ defined by*

$$K_f(x, x') := f(x^T x')$$

*is **strictly positive define** for any $n_0 \geq 1$ if and only if the coefficients $b_n > 0$ for infinitely many even and odd integer $n$.*

Now we are ready to prove the kernel or covariance function $\Sigma^L$ is strictly positive definite by using Gaussian measure techniques on the existence of positive definiteness.

**Lemma D.2.** *Suppose $\phi$ is non-polynomial Lipschitz continuous. If $\Sigma^\ell$ is strictly positive, then $\Sigma^{\ell+1}$ is also strictly positive definite.*

*Proof.* Assume the contrary. Then there exists a finite distinct collection $\{x_i\}_{i=1}^n$ and some constants $\{c_i\}_{i=1}^n$ such that

$$0 = \sum_{i,j=1}^n c_i c_j \Sigma^{\ell+1}(x_i, x_j) = \mathbb{E}\left[\sum_{i=1}^n c_i \phi(u_i)\right]^2.$$

This indicates $\sum_{i=1}^n c_i \phi(u_i) = 0$ almost surely. Note that we have the random variables $(u_i, u_j)$ follows Gaussian distribution given by

$$(u_i, u_j) \sim \mathcal{N}(0, A^\ell(x_i, x_j)).$$

WLOG, we can assume $c_1 \neq 0$. Then for some $\phi(u_1) \neq 0$, we choose $u_1 = \cdots = u_n = u_2$. Then

$$c_1 \phi(u_1) + (c_2 + \cdots + c_n)\phi(u_1) = 0,$$

indicates $c_1 = -(c_2 + \cdots + c_n)$. Then for any $u \neq u'$, we have

$$c_1 \phi(u) + (-c_1)\phi(u') = 0$$

This implies $\phi(u) = \phi(u')$, but it contradicts $\phi$ is non-constant.

$\square$

**Lemma D.3.** *Suppose $\phi$ is non-polynomial Lipschitz continuous. Then $\Sigma^2$ is strictly positive definite.*

*Proof.* For $\ell = 2$, we have

$$\Sigma^2(x, x') = \sigma_2^2 \mathbb{E}_{(u,v) \sim \mathcal{N}(0, A^1(x,x'))} [\phi(u)\phi(v)],$$

where

$$A^1(x, x') = \begin{bmatrix} 1 & \langle x, x' \rangle \\ \langle x', x \rangle & 1 \end{bmatrix}.$$

Then we have

$$\Sigma^2(x, x') = \sigma_2^2 \hat{\mu}(x^T x')$$

where $\mu(x) := \phi(x \sigma_u)$.

Clearly, $\mu$ is Lipschitz continuous since $\phi$ is. Let the expansion of $\mu$ in Hermite polynomials $\{h_n\}_{n=0}^\infty$ to be given as $\mu = \sum_{n=0}^\infty a_n h_n$. Then we can write $\hat{\mu}$ as $\hat{\mu}(\rho) = \sum_{n=0}^\infty a_n^2 \rho^n$. Then we have

$$\Sigma^2(x, x') = \sigma_w^2 \hat{\mu}(x^T x') = \sigma_w^2 \sum_{n=0}^\infty a_n^2 (x^T x')^n.$$

Since $\phi$ is assumed non-polynomials, $\mu$ is also non-polynomial, and so there are infinitely many number of nonzero $a_n$ in the expansion. Thus, $b_n := a_n^2 > 0$ for infinitely many even and odd numbers. Since $\sigma_w^2 > 0$, we have $\Sigma^2$ is strictly positive definite. $\square$

Then we obtain $\Sigma^L$ is strict positive definite by combining Lemma D.2 and D.3

# E    Proof of Lemma 4.2

This section we show the limiting covariance function $\Sigma^*$ is well defined. As each $\Sigma^L$ satisfies Cauchy-Schwartz inequality, it suffices to show $\Sigma^*(x, x)$ is well defined, which is given in Lemma E.1.

**Lemma E.1.** *Choose $\sigma_w > 0$ small for which $\beta := \frac{\sigma_w^2}{2}\mathbb{E}|z|^2|z^2 - 1| < 1$, where $z$ is standard Gaussian random variable. Then we have for every $x \in \mathbb{S}^{n_{in}-1}$ and $\ell \in [2, L]$*

$$\left|\Sigma^{\ell+1}(x,x) - \Sigma^\ell(x,x)\right| \leq \beta \left|\Sigma^\ell(x,x) - \Sigma^{\ell-1}(x,x)\right|. \tag{52}$$

*Therefore, $\Sigma^*(x,x) := \lim_{\ell\to\infty} \Sigma^\ell(x,x)$ exists uniquely and*

$$0 < \Sigma^*(x,x) \leq (1 + 1/\beta)\Sigma^2(x,x). \tag{53}$$

*Proof.* Fix $x$ and we denote $\sigma_\ell^2 := \Sigma^\ell(x,x)$ to simplify the notation. Define function $\Phi(\sigma) := \mathbb{E}_{u\sim\mathcal{N}(0,\sigma^2)}\phi(u)^2$

$$\begin{aligned}
\sigma_{\ell+1}^2 - \sigma_\ell^2 &= \sigma_w^2 \left(\mathbb{E}_{u^{\ell+1}\sim\mathcal{N}(0,\sigma_\ell^2+\sigma_1^2)}\phi(u^{\ell+1})^2 - \mathbb{E}_{u^\ell\sim\mathcal{N}(0,\sigma_{\ell-1}^2+\sigma_1^2)}\phi(u^\ell)^2\right) \\
&= \sigma_w^2 \left(\Phi\left(\sqrt{\sigma_\ell^2 + \sigma_1^2}\right) - \Phi\left(\sqrt{\sigma_{\ell-1}^2 + \sigma_1^2}\right)\right) \\
&= \sigma_w^2 \int_{\sqrt{\sigma_{\ell-1}^2+\sigma_1^2}}^{\sqrt{\sigma_\ell^2+\sigma_1^2}} \Phi'(t)dt \\
&= \sigma_w^2 \int_{\sqrt{\sigma_{\ell-1}^2+\sigma_1^2}}^{\sqrt{\sigma_\ell^2+\sigma_1^2}} \frac{1}{t}\mathbb{E}_z\phi(tz)^2(z^2 - 1)dt \\
&\leq \sigma_w^2 \int_{\sqrt{\sigma_{\ell-1}^2+\sigma_1^2}}^{\sqrt{\sigma_\ell^2+\sigma_1^2}} \frac{1}{t}\mathbb{E}_z|tz|^2\left|z^2 - 1\right|dt \\
&= \sigma_w^2 \mathbb{E}_z|z|^2\left|z^2 - 1\right| \int_{\sqrt{\sigma_{\ell-1}^2+\sigma_1^2}}^{\sqrt{\sigma_\ell^2+\sigma_1^2}} tdt \\
&= \frac{\sigma_w^2\mathbb{E}_z|z|^2\left|z^2 - 1\right|}{2}\left|\sigma_\ell^2 - \sigma_{\ell-1}^2\right| \\
&= \beta\left|\sigma_\ell^2 - \sigma_{\ell-1}^2\right|,
\end{aligned}$$

where $\beta := \frac{\sigma_w^2\mathbb{E}_z|z|^2|z^2-1|}{2}$. As we choose $\sigma_w$ small such that $\beta < 1$, then the mapping

$$\sigma_{\ell+1}^2 = \mathbb{E}_{u\sim\mathcal{N}(0,\sigma_\ell^2+\sigma_1^2)}\left[\phi(u)^2\right]$$

is a contraction. Thus, it has unique fixed point $\sigma_*$ such that

$$\sigma_*^2 = \mathbb{E}_{z\sim\mathcal{N}(0,\sigma_*^2+\sigma_1^2)}\phi(u)^2. \tag{54}$$

In addition, let $\tau_\ell^2 = \sigma_\ell^2 + \sigma_1^2$ and $\tau_1^2 = \sigma_1^2$, then we have

$$\left|\tau_{\ell+1}^2 - \tau_\ell^2\right| = \left|\sigma_{\ell+1}^2 - \sigma_\ell^2\right| \leq \beta\left|\sigma_\ell^2 - \sigma_{\ell-1}^2\right| = \beta\left|\tau_\ell^2 - \tau_{\ell-1}^2\right|.$$

Then we repeat this inequality for $\ell$ times and obtain

$$\left|\tau_{\ell+1}^2 - \tau_\ell^2\right| \leq \beta^{\ell-1}\left|\tau_2^2 - \tau_1^2\right|.$$

As LHS is $\left|\sigma_{\ell+1}^2 - \sigma_\ell^2\right|$ and RHS is $\sigma_2^2$, we obtain

$$\left|\sigma_{\ell+1}^2 - \sigma_\ell^2\right| \leq \beta^{\ell-1}\sigma_2^2.$$

Thus, we have

$$\left|\sigma_{\ell+1}^2 - \sigma_2^2\right| \leq \sum_{s=2}^\ell \left|\sigma_{s+1}^2 - \sigma_s^2\right| \leq \sum_{s=2}^\ell \beta^{s-1}\sigma_2^2 \leq \frac{1}{\beta}\sigma_2^2.$$

Therefore, we obtain

$$\sigma_\ell^2 \leq \left(1 + \frac{1}{\beta}\right)\sigma_2^2 < \infty, \quad \forall \ell \geq 2.$$

Now, suppose $\sigma_* = 0$, then we have equation

$$\begin{aligned}
0 =& \sigma_*^2 \\
=& \mathbb{E}_{u \sim \mathcal{N}(0, \sigma_*^2 + \sigma_1^2)} \phi(u)^2 \\
=& \mathbb{E}_{u \sim \mathcal{N}(0, \sigma_1^2)} \phi(u)^2 \\
=& \mathbb{E}_{u \sim \mathcal{N}(0, 1)} \phi(u)^2
\end{aligned}$$

where we use the fact $\sigma_1^2 = 1$. The equation above implies $\phi(u) = 0$ almost surely, which is impossible since $u$ follows standard Gaussian and $\phi$ is nonconstant. $\qquad\square$

### E.1 $\quad \Sigma^*(x, x) = \Sigma^*(x', x')$

In this subsection, we will first show $\Sigma^\ell(x, x) = \Sigma^\ell(x', x')$ for all $x, x'$. The desired result is obtained by letting $\ell \to \infty$.

Given $x_i$ and $x_j$, let $A_{ij}^\ell := \Sigma^\ell(x_i, x_j)$. We prove by induction. For the basic case, we have

$$A_{ii}^1 = \mathbb{E}\left|\sigma(x_i^T z)\right|^2 = \mathbb{E}\left|\sigma(u_j^1)\right|^2 = \mathbb{E}\left|\sigma(u_j^1)\right|^2 = A_{jj}^1,$$

where we use the fact $u_i \overset{i.i.d.}{\sim} \mathcal{N}(0, 1)$ due to $\|x_i\|^2 = 1$.

Assume the result holds for $\ell - 1$. Then we will show the result for $\ell$. Note that

$$\mathrm{Var}(u_i^{\ell-1}) = A_{ii}^{\ell-1} + A_{ii}^1 = A_{jj}^{\ell-1} + A_{jj}^1 = \mathrm{Var}(u_j^{\ell-1}),$$

where the last equality holds follow from the inductive hypothesis. As each $u_i^{\ell-1}$ is a centered Gaussian random variable, equal variance implies equal distribution. Then we obtain

$$A_{ii}^\ell = \mathbb{E}_{u_i^{\ell-1} \sim \mathcal{N}(0, A_{ii}^{\ell-1} + A_{ii}^1)}\left|\sigma(u_i^{\ell-1})\right|^2 = \mathbb{E}_{u_j^{\ell-1} \sim \mathcal{N}(0, A_{jj}^{\ell-1} + A_{jj}^1)}\left|\sigma(u_j^\ell)\right|^2 = A_{jj}^\ell.$$

Then let $\ell \to \infty$ and we obtain the desired result.

## F  Proof of Lemma 4.1

By choosing small $\sigma_w$, it follows from Theorem A.2 that the transition 2 becomes a contraction mapping so that we obtain the following results.

**Lemma F.1.** *Choose $\sigma_w > 0$ small for which $\gamma := 2\sqrt{2}\sigma_w < 1$. Then for every $x \in \mathbb{S}^{n_{in}-1}$ and for any $k \geq \ell$, we have*

$$\|h^\ell(x) - h^k(x)\| \leq \frac{\gamma^\ell(1 - \gamma^{k-\ell})}{1 - \gamma}\|h^1(x)\|, \quad a.s. \tag{55}$$

*Consequently, the equilibrium point $h^*(x)$ is uniquely determined a.s. Additionally, we have $\|h^\ell(x)\| \leq \frac{1-\gamma^\ell}{1-\gamma}\|h^1(x)\|$ a.s.*

To simplify the notation, let us denote

$$A_{n,\ell} := \frac{1}{n}\left\langle h^\ell(x), h^\ell(x')\right\rangle \tag{56}$$

By using Lemma F.1, we can show $\lim_{\ell \to \infty} A_{n,\ell} = B_n$ in uniformly $n$. Then the desired result is obtained by using similar argument from Moore-Osgood theorem for interchanging limits.

**Lemma F.2.** *For any $n \in \mathbb{N}$ and $k \geq \ell$, we have $|A_{n,\ell} - A_{n,k}| \leq 2(1-\gamma)^{-2}\gamma^\ell$.*

By using Lemma F.2, we can show the two iterated limits exist and equal to the double limit.

Let $\varepsilon > 0$, then there exists $L(\epsilon) \in \mathbb{N}$ such that $2(1-\gamma)^{-2}\gamma^L \le \epsilon$. Thus, $k \ge \ell \ge L(\epsilon)$ implies

$$|A_{n,\ell} - A_{n,k}| \le \epsilon.$$

Then let $n \to \infty$, it follows from Theorem 4.1 that $A_{n,\ell} \overset{a.s.}{\to} \Sigma^\ell(x,x') := \Sigma^\ell$ and so

$$\left|\Sigma^\ell - \Sigma^k\right| \le \epsilon.$$

Therefore, $\{\Sigma^\ell\}$ is a Cauchy sequence that must converge to a unique limit $\Sigma^* := \Sigma^*(x,x')$ because of the completeness of $\mathbb{R}$. Furthermore, let $k \to \infty$. we obtain

$$\left|\Sigma^\ell - \Sigma^*\right| \le \epsilon.$$

Since have $h^\ell(x) \to h^*(x)$ as $\ell \to \infty$, there exits $N(\epsilon, L) \in \mathbb{N}$ such that $n \ge N_2(\epsilon, L)$ implies

$$|A_{n,L} - H_n| \le \epsilon,$$

where $H_n := \frac{1}{n}\langle h^*(x), h^*(x')\rangle$. Combines everything together, $n \ge N$ implies

$$|H_n - \Sigma^*| \le |H_n - A_{n,L}| + \left|A_{n,L} - \Sigma^L\right| + \left|\Sigma^L - \Sigma^*\right| \le 3\varepsilon.$$

Therefore, $\lim_{n\to\infty} H_n \overset{a.s.}{=} \Sigma^* = \lim_{\ell\to\infty} \Sigma^*$. Additionally, by taking $M := \max\{L, N\}$, we can see that the limit $\Sigma^*$ equal to the double limit $\lim_{\ell,n\to\infty} A_{n,\ell}$.

## F.1 Proof of Lemma F.1

It follows from Theorem A.2 that $\frac{1}{\sqrt{n}}\|W\| \le 2\sqrt{2}\sigma_w$ a.s. Then we can choose a small $\sigma_w$ for which $\gamma := 2\sqrt{2}\sigma_w < 1$. Then for any $\ell \ge 0$, the Lipschitz continuity of $\phi$ implies

$$
\begin{aligned}
\|h^{\ell+1} - h^\ell\| =& \frac{1}{\sqrt{n}}\|\phi(Wh^\ell + g^1) - \phi(Wh^{\ell-1} + g^1)\| \\
\le& \frac{1}{\sqrt{n}}\|Wh^\ell - Wh^{\ell-1}\| \\
\le& \frac{1}{\sqrt{n}}\|W\|\|h^\ell - h^{\ell-1}\| \\
\le& \gamma\|h^\ell - h^{\ell-1}\|.
\end{aligned}
$$

Thus, we repeat this argument $\ell$ times and obtain

$$\|h^{\ell+1} - h^\ell\| \le \gamma^\ell\|h^1 - h^0\| = \gamma^\ell\|h^1\|$$

From here, for any $k \ge \ell \ge 0$, we have

$$\|h^\ell - h^k\| \le \sum_{s=\ell}^{k-1}\|h^s - h^{s+1}\| \le \sum_{s=\ell}^{k-1}\gamma^s\|h^1\| \le \frac{\gamma^\ell(1 - \gamma^{k-\ell})}{1 - \gamma}\|h^1\|. \tag{57}$$

Thus, it follows from the completeness of $\mathbb{R}^m$ that the unique $h^*(x)$ exists. Additionally, let $k \to \infty$, then we have

$$\|h^\ell - h^*\| \le \frac{\gamma^\ell}{1 - \gamma}\|h^1\|.$$

Let $\ell = 0$, then we obtain

$$\|h^k\| \le \frac{1 - \gamma^k}{1 - \gamma}\|h^1\|.$$

### F.2 Proof of Lemma F.2

WLOG, we can assume $k \geq \ell$. To simplify the notation, we further denote $h_i^\ell = h^\ell(x)$ and $h_j^\ell = h^\ell(x')$. Then we have

$$
\begin{aligned}
|A_{n,\ell} - A_{n,k}| &= \left| \frac{1}{n} \langle h_i^\ell, h_j^\ell \rangle - \frac{1}{n} \langle h_i^k, h_j^k \rangle \right| \\
&\leq \frac{1}{n} |\langle h_i^\ell, h_j^\ell \rangle - \langle h_i^\ell, h_j^k \rangle| + \frac{1}{n} |\langle h_i^\ell, h_j^k \rangle \langle h_i^k, h_j^k \rangle| \\
&\leq \frac{1}{n} \|h_i^\ell\| \cdot \|h_j^\ell - h_j^k\| + \frac{1}{n} \|h_i^\ell - h_i^k\| \cdot \|h_j^k\|
\end{aligned}
$$

It follows from Lemma F.1 that $\|h_i^\ell\| \leq \frac{1-\gamma^\ell}{1-\gamma} \|h_i^1\|$ and $\|h_j^\ell - h_j^k\| \leq \frac{\gamma^\ell}{1-\gamma} \|h_j^1\|$. Based on Theorem A.2, we have

$$
\|h_i^1\| = \|\phi(Ux_i)\| \leq \|Ux_i\| \leq C \frac{\sqrt{n}}{\sqrt{d}} \sigma_u \|x_i\|,
$$

where $C$ is some absolute fixed constant. WLOG, we can assume we choose $\sigma_u = \sqrt{d}/C$. As $\|x_i\| = 1$, we obtain

$$
\|h_i^1\| \leq \sqrt{n}, \quad \text{a.s.} \tag{58}
$$

Then we have

$$
|A_{n,\ell} - A_{n,k}| \leq \frac{2}{n} \cdot \frac{1-\gamma^\ell}{1-\gamma} \sqrt{n} \cdot \frac{\gamma^\ell}{1-\gamma} \sqrt{n} = \frac{2}{(1-\gamma)^2} \gamma^\ell
$$

## G  Proof of Theorem 4.4

By condition on the values of $h^*$, the outputs

$$
f_{\theta,k}(x) = \langle v_k, h^* \rangle
$$

are *i.i.d.* centered Gaussian random variables with covariance

$$
\hat{\Sigma}(x, x') = \langle h^*(x), h^*(x') \rangle / n.
$$

It follows from Lemma 4.1 that

$$
\hat{\Sigma}(x, x') \overset{a.s.}{\to} \Sigma^*(x, x').
$$

Specifically, the covariance $\Sigma^*$ is deterministic and hence independent to $h^*$. Consequently, the conditioned and unconditioned distributions of $f_{\theta,k}$ are equal in the limit of $n \to \infty$: they are *i.i.d.* centered Gaussian random variables with covariance $\Sigma^*$.

## H  Proof of Theorem 4.5

Equipped with the notion of dual activation and Theorem D.1, we are ready to prove Theorem 4.5, *i.e.*, $\Sigma^*$ is strictly positive definite.

By Lemma E.1, we have $\Sigma^*(x, x) = \Sigma^*(x', x') := c$ and $0 < c < \infty$ for all $x, x'$. Then we have

$$
\Sigma^*(x, x') = \mathbb{E}_{u(x), u(x') \sim \mathcal{N}(0, A^*)} \left[ \phi(u(x)) \phi(u(x')) \right]
$$

where

$$
A^* = \begin{bmatrix} \Sigma^*(x, x) + \Sigma^1(x, x) & \Sigma^*(x, x') + \Sigma^1(x, x') \\ \Sigma^*(x', x) + \Sigma^1(x', x) & \Sigma^*(x, x) + \Sigma^1(x', x') \end{bmatrix} = \begin{bmatrix} c+1 & \Sigma^*(x, x') + \langle x, x' \rangle \\ \Sigma^*(x, x') + \langle x, x' \rangle & c+1 \end{bmatrix}.
$$

By changing variable with $u(x) = \sqrt{c+1} z(x)$, we obtain

$$
\Sigma^*(x, x') = \mathbb{E} \left[ \mu(z(x)) \mu(z(x')) \right] = \hat{\mu} \left( \frac{\Sigma^*(x, x') + \langle x, x' \rangle}{c+1} \right),
$$

where $\hat{\mu} : [-1, 1] \to \mathbb{R}$ is dual activation of activation function $\mu(z) := \phi(\sqrt{c+1}z)$.

Let $\mu = \sum_{n=0}^{\infty} a_n h_n$ be the Hermite expansion, then we obtain $\hat{\mu}$ as

$$\hat{\mu}(\rho) = \sum_{n=0}^{\infty} a_n^2 \rho^n.$$

Therefore, $\Sigma^*$ has the expression

$$\Sigma^*(x, x') = \sum_{n=0}^{\infty} a_n^2 \left( \frac{\Sigma^*(x, x') + \langle x, x' \rangle}{c + 1} \right)^n.$$

Since $\phi$ is non-polynomial, so is $\mu$, and hence, there is an infinite number of nonzero $a_n$'s. By Theorem D.1, we can conclude that $\Sigma^*$ is strictly positive definite and complete the proof.

# I  Additional Experimental Results

Code is made at https://github.com/deqg/deq.git.

**Distribution of the output distribution**

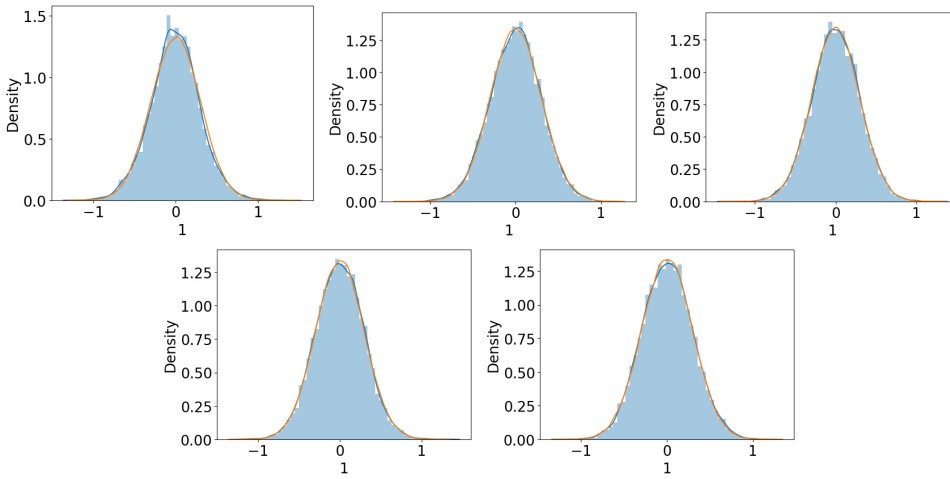

Figure 5: Histplot of the distribution of $h^L$ for five neural networks with widths 10, 50, 100, 500, 1000 (left to right); KS statistics: $0.0154, 0.0080, 0.0054, 0.0065, 0.0068$, pvalue: $0.0173, 0.5395, 0.9331, 0.7924, 0.7446$.

**Results for different activation function relu**

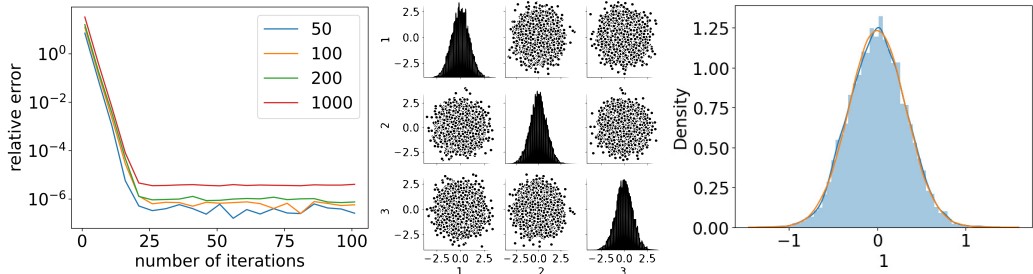

Figure 6: Plot of Figure 1 with activation function relu

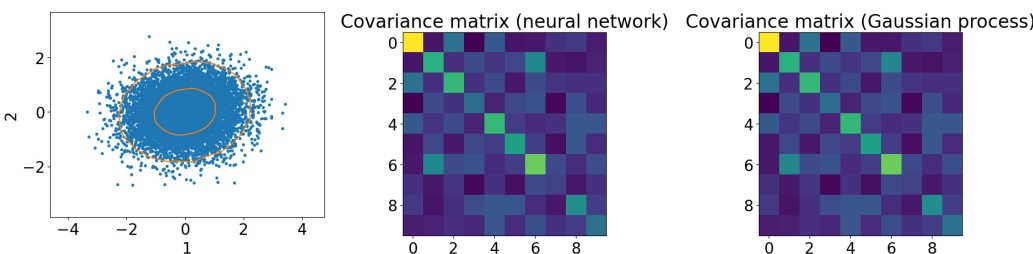

Figure 7: Plot of Figure 2 with activation function relu

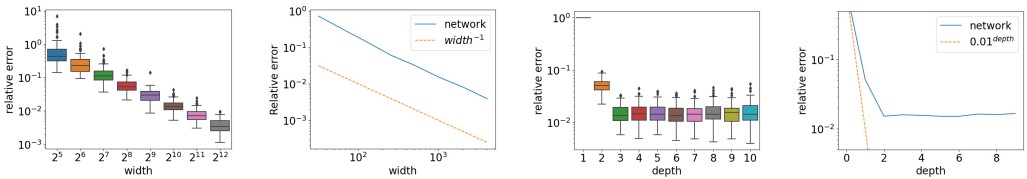

Figure 8: Plot of Figure 3 with activation function relu

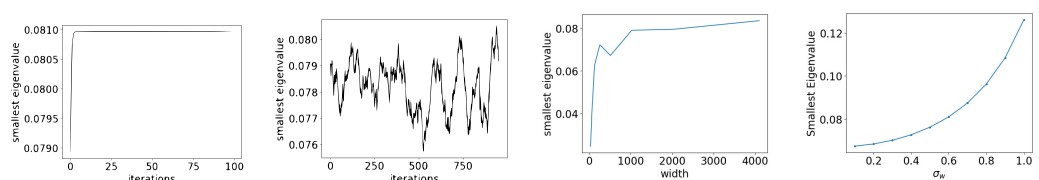

Figure 9: Plot of Figure 4 with activation function relu

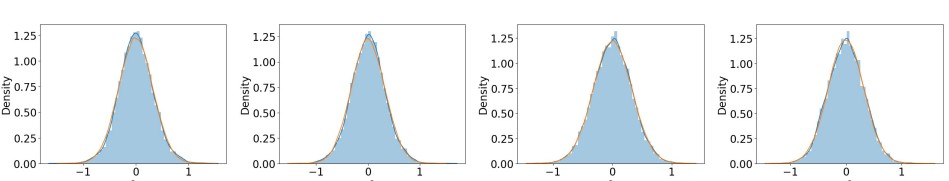

Figure 10: Plot of Figure 5 with activation function relu; KS statistics: $0.1217, 0.0137, 0.0127, 0.0070, 0.0056$, pvalue: $1.88 \times 10^{-129}, 0.0456, 0.0774, 0.7138, 0.9057$.

