# OpenReview forum: "Wide Neural Networks as Gaussian Processes: Lessons from Deep Equilibrium Models"
_NeurIPS.cc/2023/Conference — NeurIPS 2023 poster_

### Official Review · Reviewer_2cEL · 2023-06-14

**Soundness:** 4 excellent
**Presentation:** 4 excellent
**Contribution:** 3 good
**Rating:** 6
**Confidence:** 2

**Summary:**

The authors investigate the infinite-width limit of a DEQ, and prove that the output converges to a Gaussian process. Their result importantly leverages the intermediary analysis of a finite-depth, finite-width DEQ. Their main technical result is that the limit of infinite width and infinite depth commute for such networks, which they build upon to establish the convergence to a Gaussian process. Numerical checks are presented to bolster the claim.

**Strengths:**

The paper is very clearly written and easy to follow, with the main technical points being sufficiently discussed, and the relevant context being provided. Cautious numerical evidence is further provided to bolster the claims. Overall, the paper is mostly technical in nature, and altough it does not discuss the generalization properties of infinite-width DEQs, this result should be interesting to some in the NeurIPS machine learning theory community.

**Weaknesses:**

I have not read the proof, and am not familiar with the literature of DEQs, and therefore give a low confidence score. The presentation is sound and I am convinced by the numerical checks. As a very minor remark, while I do understand discussion about the generalization ability of infinite-width DEQs is out of the scope of the present work, I do feel like the inclusion of some simple empirical comparisons with other infinite-width limits of neural networks (NNGPs and NTKs) would benefit the overal reach of the work. I have a number of questions, which I list below.

**Questions:**

- The authors discuss how previous works show that for MLPs and ResNets, the infinite width and depth limits do not commute, while they show they do for DEQs. However, little discussion is provided as to why this difference arises: is it because of the share weights of the DEQ, or the input injection at each layer? I would find further intuition and comparison to MLPs helpful and insighful.

- It is not clear why $\sigma_u$ does not enter in Lemma 4.2. Naively, the $\sigma_u \to 0$ limit should correspond to a MLP, for which the two limits do not commute. Is it the case that (14) holds for any $\sigma_u>0$?

- (Minor) To my awareness, the recursions (7-13) for the infinite-width GP kernel of a DEQ are new. Could the authors provide more intuition as to how the kernel of a infinite-width but finite-depth DEQ qualitatively differs from the MLP GP kernel ($\sigma_u=0$)? For instance, a plot of the spectrum of the kernel for various $\sigma_u$ in the supplementary material would help build up intuition.

- (Minor) l73: the sentence is written twice.

**Limitations:**

The paper is purely theoretical in nature and as such does not pose any foreseeable societal impact. The technical limitations of the work a clearly stated therein.

---

> ### Author Rebuttal · Authors · 2023-08-09
>
> **Response to Question 1**: Thank you for your insightful comments and questions. Limits in DEQs can commute lies in the strategic utilization of input injection and careful selection of variance parameters. Let us provide further intuition for a clearer understanding.
>
> In DEQs, the input injection is a crucial aspect that allows us to carefully choose the covariance parameter $\sigma_w$ to control the magnitudes of $h^{\ell}$ as the depth approaches infinity. By selecting a relatively small $\sigma_w$, we ensure that the magnitude of $h^{\ell}$ remains close to that of $h^{1}$. This choice guarantees that $h^{\ell}$ converges uniformly on the width $n$ to the desired fixed point $h^{*}$.
>
> On the other hand, traditional MLPs lack the input injection mechanism, which often leads to unstable behaviors as the depth grows. The absence of input injection in MLPs can cause the preactivation vectors to become unbounded or vanish as the depth increases, resulting in a failure of the limits to commute and a loss of expressivity [13,14,8]. Similarly, even though ResNets use skip connections, they can suffer from these issues if poor scaling strategies are employed [8,9]. Therefore, both input injection (or skip connections) and appropriate scaling strategies play crucial roles in ensuring that the limits in DEQs commute.
>
> The insights gained from our study can shed light on other neural networks when considering depth becoming large. For achieving well-defined deep neural networks, it is essential to control the magnitudes of preactivation and post-activation vectors. One way to accomplish this is to add input injection (or skip connections) combined with the appropriate choice of small covariance parameters.
>
> Regarding the shared weights, it is not the critical reason for ensuring commutative limits. Instead, neural networks with shared weights, such as DEQs, are popular mainly because they can achieve competitive performance using significantly less memory storage. However, the analysis for networks with shared weights differs from those with independent weights. For instance, as demonstrated in [9], the commutative limits for ResNet with independent weights were established using results from stochastic differential equations, but these cannot be directly applied to DEQs or other neural networks with shared weights.
>
> **Response to Question 2**: Thank you for pointing that out, and we appreciate your valuable input. You are correct in identifying that we omitted the condition for $\sigma_u$, and we apologize for the oversight. To clarify, the result in Lemma 4.2 indeed holds for any fixed $\sigma_u > 0$.
>
> When $\sigma_u$ is chosen close to zero, DEQs behave like a trivial neural network. Specifically, as $\sigma_u$ approaches zero, the preactivation $g^1=Ux$ becomes very close to zero. Given the assumption that $\phi(0)=0$, the activation function output $h^1=\phi(g^1)$ will also be close to zero. As a result, in the subsequent layers, when computing $g^2=Wh^1$ and $h^2=\phi(g^2+g^1)$, both $g^2$ and $h^2$ will remain close to zero. Consequently, this DEQ effectively reduces to a trivial neural network that outputs a vector close to zero.
>
> We will update the manuscript to include this important clarification, and we thank you for helping us improve the accuracy and rigor of our work. Your feedback has been invaluable in strengthening our paper and making it more accessible to readers.
>
> **Response to Question 3**: We greatly appreciate your insights into our paper. Your suggestion regarding the qualitative distinction between the kernel of an infinite-width but finite-depth DEQ and the MLP GP kernel ($\sigma_u=0$) is valuable. To provide enhanced intuition on this aspect, we have thoughtfully incorporated plots depicting the spectrum of the kernel for various $\sigma_u$ values. These elucidating plots can be found in Figure 3 (right) within the updated PDF file featured in the "global" response section.
>
> As you delve into these plots, a discernible trend emerges. With increasing $\sigma_u$, the smallest eigenvalue of the kernel consistently exhibits a rising trajectory, a pattern observed in both theoretical analyses and simulation results. Particularly noteworthy is the scenario where $\sigma_u=0$ (or close to $0$). Here, the smallest eigenvalue of the kernel is close to zero, reflecting the diminishing pre-activation vector $g^{\ell}$ as depth grows. This phenomenon arises due to the application of a small $\sigma_w$ that ensures fixed point existence, consequently yielding a shared weight matrix $W$ satisfying $|W|:=\gamma < 1$ (with high probability). This, in turn, leads to the gradual trivialization of the covariance function or kernel.
>
> Conversely, the introduction of positive $\sigma_u$ solves this issue. Lemma~F.2 ensures the persistence of the pre-activation rather than its eventual vanishing as depth progresses. By leveraging input injection (i.e., $\sigma_u > 0$), the neural network's stability is substantially enhanced compared to scenarios devoid of this input injection.
>
> **Response to Question 4**: Thank you for pointing out the repetition in sentence l73. We will remove the duplicate sentence in the revised version of the paper. Your feedback is much appreciated.

---

> > ### Comment · Reviewer_2cEL · 2023-08-11
> > **Acknowledgements**
> >
> > I thank the authors for the clarification, and do not change my scoring.

---

### Official Review · Reviewer_A2iv · 2023-06-26

**Soundness:** 3 good
**Presentation:** 2 fair
**Contribution:** 2 fair
**Rating:** 5
**Confidence:** 3

**Summary:**

This paper focuses on the DEQ (deep equilibrium) model, an infinitely deep neural network with shared weight matrices across layers. The authors show that the network approaches a Gaussian process as the network width goes to infinity, and the limit of infinite width and infinite depth commute, also, the Gaussian vectors remain non-degenerate as network depth tends to infinity for any pairs of distinct inputs. These results do not hold for previously well-studied wide neural networks without shared weights.

**Strengths:**

The paper presents meticulous analysis on the infinite width and infinite depth limits of the DEQ model (networks with tied weights) and specifically the rate of convergence of the two limits. The theoretical analysis is supported by numerical results.

**Weaknesses:**

1. While technically sound, it is unclear what are the potential insights and contribution of this work to the field. I recommend that the authors add a paragraph in the conclusion section (or if possible include some numerics) on the potential implications of their main results (that the infinite depth and width limits commute for DEQ and that the structure of the limiting kernel is preserved). For example, does this explain why DEQ’s achieve competitive results with the state-of-the-art? How does the result shed light on the infinite width limit of vanilla RNNs?
2. This work focuses on properties of the DEQ at initialization, and there is no learning in the network.

**Questions:**

1. I am not familiar with the line of work on DEQs, but based on the references I think it refers to the method of computing the fixed point of the infinitely deep networks with tied-weights, whereas in the paper it simply refers to the network structure?
2. Line 91: What are the conditions on A here (elements in A need to be subgaussian?)
3. Theorem 3.1: Here you introduce the covariance function, however, it may be useful to point out that the covariance function here is different than the covariance of f in NNGP and NTK, in NNGP the readout weights of the network is learned whereas in NTK all the weights are learned. If I understand correctly, here the covariance function is simply f at initialization with all the weights being Gaussian. As you also mention NTK in line 129 it may be helpful to stress the distinction.
4. Fig 4: why is there a large jump at the initial stage in the theory but not in simulation for the smallest eigenvalue over iterations?
5. Fig 4 the last panel: can you confirm this result with simulations?

**Limitations:**

The authors should add a paragraph on limitations in the conclusion section.

---

> ### Author Rebuttal · Authors · 2023-08-09
>
> **Response to Weakness 1**: Thank you for your input. We establish the NNGP correspondence for DEQs, shedding light on why they can compete with advanced models. This is due to their tendency to exhibit Gaussian behavior, when the width is large.
>
> One implication is its potential application to Gaussian processes. To explore this, we conducted experiments, comparing GPs utilizing the NNGP kernel with trained DEQs of varying widths on real datasets. As depicted in Figure~1 of the updated PDF in the "global" response, we observe that GPs with NNGP kernel outperform finite-width trained DEQs, and DEQs' performance converges to NNGP as width increases, akin to feedforward networks [12].
>
> Moreover, Our methodology can yield similar outcomes as in [19,1], establishing NNGP correspondence for RNNs. Moreover, our analysis can extend to broader networks, especially deep ones, as we successfully demonstrate limit commutativity and present a new approach to prove strictly positive definiteness of NNGP kernel.
>
> **Response to Weakness 2**: Thank you for your valuable comments. Our current focus is establishing NNGP correspondence for DEQs, not training analysis. However, our findings offer a foundation for studying training dynamics in both infinite and finite widths.
>
> In [10], the training dynamics are described via an ordinary differential equation governed by the NTK. [10] also underscores the NNGP kernel is part of NTK and strictly positive NTK determines convergence of training process in both finite-width [5] and infinite-width [10]. Hence, the strictly positive definiteness of NNGP kernel established in Theorem 4.5 significantly determines training behavior and stability across infinite and finite widths.
>
> Additionally, we have conducted experiments comparing NNGP predictions to trained DEQs with varying widths on real datasets. The results are depicted in the PDF file of the "global" response. Figure 1 illustrates that the predictions of trained DEQs increase and converge to NNGP predictions as the width increases, akin to feedforward networks [12].
>
> **Response to Question 1**: Thank you for your question, and we appreciate the opportunity to clarify the concept of Deep Equilibrium Models (DEQs).
>
> As introduced in [3], DEQs represent a distinctive neural network structure, whose latent feature vector $h^*$ is provided implicitly as the limit of the post-activation vector $h^{\ell}$ when $\ell\rightarrow\infty$. This characteristic categorizes DEQs as an infinite-depth neural network with shared weights.
>
> Furthermore, $h^*$ can be interpreted as a fixed or equilibrium point within the equilibrium equation, as illustrated in Equation (4). Notably, the equilibrium equation can be much more complex in practice [3]. The fixed point can be computed through iterative transitions or efficient root-finding techniques such as Newton-like methods [3] and Anderson acceleration [17]. Remarkably, our results apply to the DEQs, regardless of how one computes the fixed point.
>
> **Response to Question 2**: Thank you for your question. As indicated in Theorem A.1 and Theorem A.2 from Appendix A, we assume that $A_{ij}$ are i.i.d. standard Gaussian. This assumption aligns with our random initialization introduced in Equation (5).
>
> However, it is worth noting that the results can be extended to subgaussian with some level of dependence for rectangular matrices. For more detailed and advanced results, we recommend referring to the works [15,16].
>
> **Response to Question 3**: Thank you for your question, and we appreciate the opportunity to clarify the distinction between the covariance function in our paper and its relation to NNGP and NTK.
>
> In the infinite-width limit, neural network with random weights is equivalence to a Gaussian process with specific kernels, known as the NNGP correspondence [10,12]. These kernels are utilized in Bayesian inference or Support Vector Machines, yielding results comparable to trained neural networks trained [12]. Our work shows that DEQs also exhibit this NNGP correspondence, as demonstrated in Theorem 4.4, where the NNGP kernel is the covariance function $\Sigma^*$ defined in Theorem 4.1 and Lemma 4.1.
>
> Conversely, in the same limit, [10] illustrates that the dynamics of neural networks under training can be described as an ordinary differential equation governed by another kernel called NTK. Notably, the NTK is distinct from but related to the NNGP kernel. For instance, a strictly positive NNGP kernel implies a strictly positive NTK, but not vice versa [10].
>
> **Response to Question 4**: Thank you for raising this matter, and we apologize for any confusion caused by the misleading labels and captions in Figure 4. To clarify, the second subgifure in Figure 4 illustrates the distribution of $\lambda_{\min}(K^*)$ across 1000 networks instead of $\lambda_{\min}(K^{\ell})$ through simulation. In response to this issue, we have redrawn Figure 4, and the revised plots, now denoted as Figure 3 (left) in the updated PDF file of "global" response, offer a more accurate portrayal. These updated plots demonstrate a consistent alignment between theory and simulation. Notably, both theory and simulation curves exhibit a big jump at the initial stage. That is because $K^1 = \sigma_u^2 XX^T/n_{in}$ is generally degenerated due to dependence within training data, while $K^{\ell}$ is non-degenerate for all $\ell\geq 2$ due to the nonlinear activation $\phi$.
>
> **Response to Question 5**: Certainly, we have validated this outcome through simulations. The corresponding plots are now available in the updated PDF file within the "global" response section. Specifically, Figure 3 (middle) demonstrates a clear and consistent increase in the smallest eigenvalue of $K^*$ as $\sigma_w$ increases, mirroring the theoretical computations.

---

> > ### Comment · Reviewer_A2iv · 2023-08-19
> > **Reply to rebuttal**
> >
> > Sorry for the late response. I have read the rebuttal and I appreciate the clarification and new numerical validations. Correspondingly I would like to raise my score to 5. I do find the result that the limits commute very interesting, my concern is about the DEQ model itself. It does not seem to be a commonly used model and it is vague how the results may be applicable to other related more practical architectures.

---

> > > ### Author Response · Authors · 2023-08-19
> > >
> > > Thank you for your response and for revising your score to 5. I appreciate your feedback and the recognition of the clarification and additional numerical validations provided in our rebuttal.
> > >
> > > We understand your concern about practical applications, and we are committed to exploring how our findings can be applied in more common architectures in the future.
> > > Your insights are valuable, and we appreciate your support.
> > >
> > > Sincerely,
> > >
> > > The Authors

---

### Official Review · Reviewer_bC2e · 2023-07-01

**Soundness:** 3 good
**Presentation:** 2 fair
**Contribution:** 3 good
**Rating:** 7
**Confidence:** 4

**Summary:**

This paper examines the infinite width behaviour of DEQs, a kind of neural network architecture that can be viewed as an infinite-depth RNN. They show that contrary to regular MLPs, the limit of infinite width and infinite depth in DEQs commutes. They back their theory up with some numerical experiments.

**Strengths:**

- The authors ask an interesting theoretical question and derive a surprising result (that the limits commute).
- Paper is generally well written

**Weaknesses:**

- The central importance of the commutation of the limits could perhaps be better argued for. As it stands, this seems interesting to me theoretically, but I don't see any major practical advances stemming from this result. It would be good to know if the authors see things differently.
- I found some of the Figures hard to understand, particularly Fig 3 and Fig 4. Would appreciate more detail here.


**Questions:**

- What is happening in the rightmost figure in Fig 3? It looks like there is an orange curve that goes all the way to the bottom, but the blue curve does not stay with it. I'm having trouble understanding the significance of this.
- Why does the relative error bottom out at different values in Fig 1, left?
- What is the legend in Fig 1 left? Is it width of the network?

**Limitations:**

Yes

---

> ### Author Rebuttal · Authors · 2023-08-09
>
> **Response to Weakness 1**: We appreciate the reviewer's insightful comments. It is widely recognized in the most recent literature [12,19,10] that as the width of neural networks approaches infinity, they exhibit Gaussian behavior, a phenomenon known as the Neural Network and Gaussian Process (NNGP) correspondence. This understanding has led to the successful application of NNGP kernels to Gaussian Processes (GPs), resulting in remarkable performance on real-world datasets [12,19]. Furthermore, the spectrum of the NNGP kernel plays a pivotal role in determining the global convergence of gradient-based training methods for neural networks [10,5].
>
> However, it is important to note that this NNGP correspondence cannot be guaranteed in cases where the depth of neural networks becomes significantly large. For instance, as highlighted by [9], standard feedforward networks display heavy-tail behavior instead of Gaussian behavior when depth outpaces width in convergence. To address this challenge, we employ the input injection in DEQs and carefully select variance parameters $\sigma_w$ to ensure the commutation of depth and width limits within DEQs. This allows us to establish the NNGP correspondence for DEQs. Additionally, we also demonstrate the strict positivity of the NNGP kernel. These findings contribute fundamental insights for future studies focused on the training and generalization of DEQs.
>
> To complement our theoretical findings, a new experiment is added using the NNGP for regression on the MNIST dataset, the result is provided in Figure 1 in the PDF file of the "global response". We can see from Figure 1 that NNGP outperformances trained finite-width DEQs, but the performance of DEQs tends to converge to NNGP as the width increases.
>
> **Response to Weakness 2**: Thank you for your feedback and apologize for any confusion caused by the figures. To address this concern, we have redrawn both Figure 3 and Figure 4 to ensure improved clarity. The updated figures and detailed captions are now available in the PDF file provided in the "global" response section. Additionally, we also provide detailed answers to your follow-up questions to ensure a better understanding of the figures and the underlying concepts.
>
> **Response to Question 1**: Thank you for pointing out this observation. The rightmost figure in Figure~3 depicts the convergence of the relative error $\|H^{\ell}H^{\ell}/n-K^*\|/\|K^*\|$ as the depth $\ell$ increases. The orange curve represents the fitted convergence behavior of the relative error, which is an exponential function denoted as $\gamma^{\ell}$.
>
> The reason why the blue curve does not follow the orange curve to the bottom is that the relative error is not only influenced by the depth but also by the width of the neural network. The error introduced by the width contributes to the deviation of the blue curve from the orange curve.
>
> Considering the potential for confusion and misinterpretation arising from this width-related effect, we removed the orange curve from our updated plots, which aims to provide a clearer and more precise visualization of the convergence behavior. For your convenience, we have included the revised plots in the updated PDF file accessible through the "global" response section.
>
> **Response to Question 2**: Thank you for bringing this observation. The variation in the bottoming-out values within Fig 1, left, is influenced by several factors.
>
> Primarily, the discrepancies can be understood through the lens of Lemma~F.2, wherein the relative error $\|h^{\ell+1}-h^{\ell}\|\leq \gamma^{\ell} \|h^1\|$ holds true. Incorporating Theorem A.1 alongside the Lipschitz continuity of $\phi$, we deduce that $\|h^1\|=\|\phi(Ux)\|= \mathcal{O}(\sqrt{n})$. Consequently, it follows that the relative error $\|h^{\ell+1}-h^{\ell}\|= \mathcal{O}(\gamma^{\ell}\sqrt{n})$, implying that the error would exhibit relatively greater magnitudes as the width values increase.
>
> Furthermore, it's noteworthy to note that practical considerations such as numerical computational errors can contribute to the disparities and fluctuations observed in the relative error values. These nuances might, in part, account for the diversities observed in the plot.
>
> To rectify the confusion stemming from the plot, we have undertaken enhancements in the new depiction. By incorporating more intermediary width selections (e.g., $[50,100,200,400,800,1000,2000]$), the revised plot presents a clearer and more plausible portrayal of the data. Additionally, to mitigate the influence of width $n$, we have reformulated the plot using fresh relative errors $\|h^{\ell+1}-h^{\ell}\|/\|h^{\ell}\|$. The result is a smoother and more consistent depiction of the convergence behavior for different widths, as shown in the updated PDF file's right figure within Figure 2.
>
> **Response to Question 3**: Your understanding is accurate. The legend accompanying the left side of Figure~1 denotes the width of the network. We have taken steps to refine the presentation. The new plot features revised legend labels following the format "width xxx." This adjustment provides a clearer and more intuitive grasp of the width values, avoiding confusion. We wholeheartedly value your perceptive observation and your valuable contribution.

---

> > ### Comment · Reviewer_bC2e · 2023-08-17
> >
> > Thanks for your responses - and apologies for the very late reply.
> >
> > On my first point - I think the authors do a good job outlining the theoretical significance of their work, in that it's very interesting that these limits commute, and not expected a priori. I was wondering whether the authors in addition foresee a practical advance stemming from this theoretical insight. Not having one is not a deal-breaker, but I was curious.
> >
> > Thank you very much for improving and clarifying the figures. I think this is an improvement to the paper. However, I am planning to maintain my score - I think this paper should be accepted, but I do not think this result on its own qualifies as "excellent impact". I think it is solid but I am not sure if the impact of this work will be very large, given that DEQs are not a very commonly used model, and it is unclear what the practical impact of this particular result will be.
> >
> > I am increasing my confidence score to a 4 as I'm confident this paper should be accepted.

---

### Official Review · Reviewer_tXNv · 2023-07-06

**Soundness:** 3 good
**Presentation:** 3 good
**Contribution:** 2 fair
**Rating:** 6
**Confidence:** 3

**Summary:**

This paper takes the theoretical tools for infinite width limits of fully connected deep neural networks (e.g. NNGP limits, tensor programs etc), and applies it to a deep equilibrium-type neural network. This model is seen as the depth \to infinity limit of a feedforward-type network.  Convergence to a Gaussian process is proven for this model, and the typical type of recurrence relation for the covariance function is obtained. The theoretical results are validated with several experiments that confirm the findings, and shed light on how quick the convergence to the fixed point is.

**Strengths:**

* The paper is well written and easy to follow. The proofs are explained well and seem to be free of any major errors.
* The experiments do a good job validating the theory. They cover a broad range of possible questions one might ask about the model.
* The fact that the infinite depth limit exists in a non-degenerate way is a nice finding (as opposed to other types of feedforward networks where the infinite depth limit may be degenerate in some way). This means that the idea of feeding the input $x$ into each layer directly may be a stabilizing force that can be used for very deep networks.

**Weaknesses:**

* The main result of this paper is largely what one would expect using the infinite width technology, that is to say there is no "surprises" that happen along the way. That is not to say that something is lacking in the methods applied here, just that the result is more or less what you would expect. To really make this an outstanding paper, it would be nice to additionally see some explanation of why the resulting model is interesting...for example is there some way in which it outperforms ordinary feedforward nets or resnets or is there a way in which it is more efficient. Perhaps this kind of thing appears in the existing literature on deep equilibrium models (I am not an expert on this type of model)
* Usually the big strength of the infinite width limits is that the NTK is a constant which enables a theoretical understanding of how training happens, not just initialization. This analysis of the training kernel is not carried out in this work.
* The fact that the limit with or without shared weights is identical suggests that some important theoretical considerations may be missing here...surely there is some kind of important difference between these models that this infinite width limit is not capturing. In particular, I think the training kernel for the model with or without shared weights would be different even though the conjugate kernels studied here are the same. It would be interesting to compare/contrast these more, either by experiments or by more theory to get a handle on what this model is (or isn't) actually doing.


**Questions:**

* Have you thought about the training kernel (NTK) for this model? Is there a reason the usual analysis would not go through?
* Is there any analysis on the differences between the DEQ and the standard feedforward model in the wide width limit? What are the advantages/disadvantages? This may be found somewhere in the existing literature, but this was not clear to me and the fact that this DEQ model would have some advantages would be an important part of motivating the work done here.

**Limitations:**

* One limitation that always comes up in these limiting type things is the question of how large real neural networks have to be for the theory to actually work. Some discussion or experiments specifically addressing this (e.g. showing the error in the predictions as a function of network size) could help explicitly address this.
* Related to above: It would also be interesting to understand the fluctuations around the infinite width limit and how much they effect things (which may be quite different for this model vs for ordinary feedforward networks), but that is likely beyond the scope of this paper.

---

> ### Author Rebuttal · Authors · 2023-08-09
>
> **Response to Weakness 1**: Thank you for your valuable review. We appreciate your interest in the efficiency of DEQs compared to other advanced neural networks, and your consideration of the nature of our main result.
>
> Implicit networks [1], such as DEQs, have recently gained significant attention for their versatility, encompassing models like MLPs, convolutional nets, ResNets, and RNNs [1,2]. Moreover, DEQs have also stood out for their competitive performance while economizing computational resources [2,4].
>
> In terms of expected outcomes, it's important to note that unlike shallower networks, deep networks can show instability as their depth increases. For instance, as noted in [5], feedforward networks may display heavy-tailed distributions rather than Gaussian behavior when depth converges faster than width. This instability can subsequently impact network expressivity [7,8].
>
> To address these challenges, we utilize input injection in DEQs and carefully select variance parameters $\sigma_w$ to ensure the commutativity of the two limits. This forms the basis for establishing a significant NNGP correspondence for DEQs. Our strategies and mathematical tools could potentially extend to other networks, particularly those with substantial depth.
>
> **Response to Weakness 2**: Thank you for your valuable comments. You are correct in noting that our current work does not encompass an analysis of the training, since the primary focus of our paper is to establish the NNGP correspondence to DEQs. However, our findings offer a solid foundation for probing training dynamics in scenarios of both infinite and finite widths.
>
> Furthermore, as highlighted in [3], the dynamics of networks under training can be described through an ordinary differential equation governed by the NTK. [3] also underscores the NNGP kernel is part of NTK and strictly positive NTK determines convergence of training process in both finite-width [6] and infinite-width [3] scenarios. Therefore, the strictly positive definiteness of NNGP kernel established in Theorem 4.5 significantly influences training behavior and stability across infinite and finite widths. Hence, we intend to explore training dynamics of DEQs in our future research.
>
> **Response to Weakness 3**: Thank you for your valuable response. While it holds true that the identical Gaussian process is attained in the limit regardless of shared weights, as noted in Remark 4.2 and other literature [9,10], the absence of shared weights leads to a non correlation across layers.
>
> Regarding the training kernel or NTK, you correctly identify the distinction between the NTK with and without shared weights. However, it is crucial to emphasize that the central thrust of our paper is to establish the NNGP correspondence for DEQs. In the context of finite-depth neural networks, akin to our current study, a line of studies has primarily focused on scrutinizing the NNGPs across diverse network architectures, rather than delving into the realm of NTK[10,11,5]. While training dynamics and generalization are undeniably significant, we consider it a promising avenue for future investigation due to the time limit.
>
> **Response to Question 1**: Thank you for your insightful question. The analysis of the training kernel (NTK) is indeed part of our future work. The analysis of the training kernel for DEQs could encounter analogous challenges to those encountered in analyzing the NNGP kernel for DEQs. Specifically, the training kernel may not be well-defined if the two limits (infinite width and depth) do not commute. As DEQs are defined as infinite-depth neural networks, it is crucial for the two limits to commute. However, we are optimistic that the methodologies and mathematical frameworks in this paper can serve to ensure the commutativity of these two limits, thereby rendering the training kernel or NTK well-posed.
>
> **Response to Question 2**: Thanks for your questions. The main motivation behind using DEQs lies in their practical benefits. Existing literature has highlighted that DEQs can attain competitive performance compared to other networks while demanding significantly fewer resources [2,4]. However, the theoretical analysis of DEQs is still an ongoing field of exploration. Limited research has delved into the well-posedness and training process of DEQs [12,13,14]. We intend to investigate the training and generalization ability of DEQs as part of our future work.
>
> **Response to Limitation 1**: Thank you for your valuable feedback. While this paper primarily focuses on asymptotic analysis to establish the NNGP correspondence for DEQs, addressing the practical applicability of the theoretical findings requires non-asymptotic analysis [6,15].
>
> To complement our theoretical analysis, we have conducted experiments comparing NNGP predictions with trained DEQs of varying widths on realistic datasets. The results are depicted in the updated PDF file in the "global" response section. Figure 1 illustrates that the test accuracy of trained DEQs increases and converges towards that of NNGP as the width increases. This trend aligns with observations made in the context of feedforward networks [11].
>
> **Response to Limitation 2**: We appreciate your insights. While fluctuations around the infinite width limit are interesting, our study centers on depth-width interaction and NNGP correspondence. We value your suggestions for future work and plan to consider them in subsequent research.

---

> > ### Comment · Reviewer_tXNv · 2023-08-15
> > **Acknowledgement**
> >
> > I acknowledge the rebuttal from the authors.

---

### Author Rebuttal · Authors · 2023-08-09

Dear Reviewers,

We sincerely appreciate your thorough review of our paper and your valuable comments and suggestions. We have carefully considered each of your points and have addressed them in our separate responses to your questions.

In light of your input, we have taken significant steps to enhance the clarity and comprehensibility of our work. Specifically, we have conducted new experiments that involve a comparison between NNGP predictions and trained DEQs of varying widths on real datasets. These results, along with updated plots, have been incorporated into the uploaded PDF file.

Acknowledging the confusion caused by misleading labels and captions in certain plots, we have revisited these figures, refining the labels and captions to achieve heightened clarity. Our goal is to ensure that these figures accurately represent our findings, eliminating any potential ambiguity. The revised figures are now included within the updated PDF file.

For your convenience, we have also included references in this global response that were cited in our rebuttal. This addition aims to provide you with a seamless understanding of the context to which we refer.

We deeply appreciate your time and commitment to appraising our manuscript. Your insights have been instrumental in refining our work.

Best regards,

The Authors

===

**References**

[1] Sina Alemohammad, Zichao Wang, Randall Balestriero, and Richard Baraniuk. The recurrent neural tangent kernel. arXiv preprint arXiv:2006.10246, 2020.

[2] Zeyuan Allen-Zhu, Yuanzhi Li, and Zhao Song. A convergence theory for deep learning via over-parameterization. In International conference on machine learning, pages 242–252. PMLR, 2019.

[3] Shaojie Bai, J Zico Kolter, and Vladlen Koltun. Deep equilibrium models. Advances in Neural Information Processing Systems, 32, 2019.

[4] Mostafa Dehghani, Stephan Gouws, Oriol Vinyals, Jakob Uszkoreit, and Lukasz Kaiser. Universal transformers. In International Conference on Learning Representations, 2018.

[5] Simon Du, Jason Lee, Haochuan Li, Liwei Wang, and Xiyu Zhai. Gradient descent finds global minima of deep neural networks. In International conference on machine learning, pages 1675–1685. PMLR, 2019.

[6] Laurent El Ghaoui, Fangda Gu, Bertrand Travacca, Armin Askari, and Alicia Tsai. Implicit deep learning. SIAM Journal on Mathematics of
Data Science, 3(3):930–958, 2021.

[7] Tianxiang Gao, Hailiang Liu, Jia Liu, Hridesh Rajan, and Hongyang Gao. A global convergence theory for deep relu implicit networks via
over-parameterization. In International Conference on Learning Representations, 2021.

[8] Soufiane Hayou, Arnaud Doucet, and Judith Rousseau. On the impact of the activation function on deep neural networks training. In
International conference on machine learning, pages 2672–2680. PMLR, 2019.

[9] Soufiane Hayou and Greg Yang. Width and depth limits commute in residual networks. arXiv preprint arXiv:2302.00453, 2023.

[10] Arthur Jacot, Franck Gabriel, and Cl ́ement Hongler. Neural tangent kernel: Convergence and generalization in neural networks. Advances in neural information processing systems, 31, 2018.

[11] Kenji Kawaguchi. On the theory of implicit deep learning: Global convergence with implicit layers. In International Conference on Learning Representations, 2020.

[12] Jaehoon Lee, Yasaman Bahri, Roman Novak, Samuel S Schoenholz, Jeffrey Pennington, and Jascha Sohl-Dickstein. Deep neural networks as gaussian processes. In International Conference on Learning Representations.

[13] Ben Poole, Subhaneil Lahiri, Maithra Raghu, Jascha Sohl-Dickstein, and Surya Ganguli. Exponential expressivity in deep neural networks through transient chaos. Advances in neural information processing systems, 29, 2016.

[14] Samuel S Schoenholz, Justin Gilmer, Surya Ganguli, and Jascha Sohl-Dickstein. Deep information propagation. In International Conference on Learning Representations.

[15] Terence Tao. Topics in random matrix theory, volume 132. American Mathematical Soc., 2012.

[16] Roman Vershynin. High-dimensional probability: An introduction with applications in data science, volume 47. Cambridge university press, 2018.

[17] Homer F Walker and Peng Ni. Anderson acceleration for fixed-point iterations. SIAM Journal on Numerical Analysis, 49(4):1715–1735, 2011.

[18] Ezra Winston and J Zico Kolter. Monotone operator equilibrium networks. Advances in neural information processing systems, 33:10718–10728, 2020.

[19] Greg Yang. Wide feedforward or recurrent neural networks of any architecture are gaussian processes. Advances in Neural Information
Processing Systems, 32, 2019.

---

### Comment · Area_Chair_rKkc · 2023-08-18

Dear authors, thank you for your rebuttal. I will take into consideration your unanswered responses about significance and the focus on DEQs at initialization.

---

> ### Author Response · Authors · 2023-08-19
>
> Dear Area Chair rKkc,
>
> Thank you for your prompt response. We sincerely appreciate your understanding, and we will remain vigilant for any further updates or feedback.
>
> Best regards,
> The Authors

---

### Decision · Program_Chairs · 2023-09-21

**Decision:**

Accept (poster)

**Comment:**

This paper explores the infinite-width behavior of deep equilibrium neural networks, and finds a limiting Gaussian process behavior that holds even when depth and width limits are interchanged. Overall, the work is technically solid. Although DEQ models are not in widespread use, the insights from this work will be of interest to the NeurIPS community and therefore I recommend acceptance.

There were some concerns about the significance of the results. First, the authors’ analysis only focuses on initialization (GP limit) and not training (NTK limit), and the authors could do a better job of explaining the significance of interchanging width/depth limits. The authors adequately addressed these concerns in their rebuttal, leading to consensus that this work is a valuable contribution to the analysis of infinite width limits.